# Synergistic Antitumor Effects of Caerin Peptides and Dendritic Cell Vaccines in a 4T-1 Murine Breast Cancer Model

**DOI:** 10.3390/vaccines13060577

**Published:** 2025-05-28

**Authors:** Rongmi Mo, Junjie Li, Xinyi Song, Jiawei Fu, Mengqi Liu, Yuandong Luo, Quanlan Fu, Jinyi Wu, Hongyin Wu, Yongxin Liang, Tianfang Wang, Xiaosong Liu, Guoying Ni

**Affiliations:** 1The First Affiliated Hospital, Clinical Medical School, Guangdong Pharmaceutical University, Guangzhou 510080, China; 15278097695@163.com (R.M.); sxy824721567@163.com (X.S.); 15142169340@163.com (J.F.); m17862862250@163.com (J.W.); 13430237300@163.com (H.W.); xiaosongl@yahoo.com (X.L.); 2Zhongao Biomedical Technology (Guangdong) Co., Ltd., Zhongshan 528400, China; kathars991230@outlook.com (J.L.); lyd69515@163.com (Y.L.); qlf2300@163.com (Q.F.); 3Medical College, Guizhou University, Guiyang 550025, China; liu7151338@163.com; 4Cancer Research Institute, Foshan First People’s Hospital, Foshan 528000, China; lyx1627066113@126.com; 5Centre for Bioinnovation, University of the Sunshine Coast, Maroochydore, QLD 4558, Australia; twang@usc.edu.au; 6School of Science, Technology and Engineering, University of the Sunshine Coast, Maroochydore, QLD 4558, Australia

**Keywords:** caerin 1.1/1.9, DC vaccine, TME, DLNs, 4T-1

## Abstract

**Background/Objectives**: Breast cancer remains a leading cause of cancer-related mortality among women worldwide, necessitating novel therapeutic strategies. This study aimed to investigate the synergistic antitumor effects of caerin peptides (F1/F3) combined with dendritic cell (DC) vaccines in a 4T-1 murine breast cancer model, providing new insights for breast cancer immunotherapy. **Methods**: In vitro experiments evaluated the effects of F1/F3 on 4T-1 cell proliferation and apoptosis. A 4T-1 breast cancer mouse model was established, and treatments included F1/F3 alone, DC vaccines (DCV_1_: loaded with whole tumor antigens; DCV_2_: loaded with F1/F3-induced apoptotic antigens), or combination therapy. Flow cytometry analyzed immune cell subsets in the tumor microenvironment and lymph nodes, while ELISA measured cytokine levels. **Results**: F1/F3 significantly inhibited 4T-1 cell proliferation and induced apoptosis while suppressing tumor growth and lung metastasis in vivo. Flow cytometry revealed increased infiltration of CD4^+^ T cells and cDC_1_ in tumors, along with reduced PD-L1 expression. DCV_2_ exhibited stronger T-cell proliferation induction and lower IL-10 secretion in vitro. Combination therapy with DCV_2_ and F1/F3 demonstrated superior tumor suppression compared to monotherapy. **Conclusions**: F1/F3 enhances antitumor immunity by modulating the tumor microenvironment, and its combination with DCV_2_ yields synergistic effects. This study provides experimental evidence for combination immunotherapy in breast cancer, with potential for further optimization of DC vaccine design to improve efficacy.

## 1. Introduction

Breast cancer is the second leading cause of cancer-related deaths among women worldwide. Despite significant advancements in early detection and treatment, its incidence continues to rise. The heterogeneity of breast cancer poses substantial challenges for its clinical management and treatment [1,2]. To date, surgery, chemotherapy, and radiotherapy remain the primary therapeutic approaches for breast cancer [3,4,5]. However, poor postoperative prognosis, high recurrence rates, and treatment resistance have limited their efficacy [6,7,8,9]. This underscores the urgent need for novel therapeutic strategies to improve outcomes in breast cancer patients.

Immunotherapy has emerged as a promising approach for cancer treatment, now becoming the standard of care for many malignancies, such as lung cancer and melanoma, as well as specific subtypes of triple-negative breast cancer (TNBC) [10,11]. Immune checkpoint inhibitors targeting the PD-1/PD-L1 pathway, such as pembrolizumab and atezolizumab, have been approved for the treatment of solid tumors, including breast cancer [12,13]. However, most TNBC patients exhibit limited responses to monotherapy with these agents, highlighting the need for combination therapies to enhance therapeutic efficacy [14,15].

Dendritic cells (DCs), as the most potent professional antigen-presenting cells in vivo, play a central role in initiating and modulating innate and adaptive immunity within the lymph nodes and the tumor microenvironment (TME) [16,17]. DC vaccines leverage the antigen-presenting capabilities of DCs by loading them with tumor-associated antigens (TAAs) and promoting their maturation ex vivo before autologous re-infusion into patients. This approach bypasses the immunosuppressive TME, generating TAA-specific cytotoxic T lymphocytes (CTLs) to kill tumor cells and induce long-term immune memory [18]. DC vaccines have been widely investigated in cancers such as non-small-cell lung cancer (NSCLC), ovarian cancer, prostate cancer, melanoma, renal cell carcinoma, and glioblastoma [19]. However, clinical applications of DC vaccines in breast cancer remain limited. Preliminary clinical trials have demonstrated the potential of DC vaccines in reducing recurrence rates. For example, an oxidized mannan-MUC1 vaccine significantly reduced recurrence in 31 breast cancer patients [20]. Moreover, a recent phase I/II trial (NCT02061332) involving 58 HER2-positive ductal carcinoma in situ (DCIS) patients validated the feasibility of DC-based vaccination and assessed different DC delivery methods [21]. Despite these promising results, the immunosuppressive nature of the TME often hinders the clinical efficacy of DC vaccines. Current research focuses on unraveling and counteracting the complex interactions within the TME that impair immune activation and effective tumor cell killing [22].

Caerin peptides, derived from the skin secretions of Australian tree frogs, are natural host defense peptides with potent anticancer and antimicrobial activities [23]. Caerin 1.1 has demonstrated cytotoxic effects against various cancer cells [24], while caerin 1.9 exhibits strong antibacterial activity [25]. Recent studies have shown that the combination of caerin 1.1 and 1.9 can inhibit cervical cancer cell proliferation in vitro and induce apoptosis via activation of the TNFα signaling pathway [26]. In vivo, caerin 1.1/1.9 suppresses the growth of HPV-positive TC-1 tumors, recruits more CD8^+^ T cells and NK cells to the tumor site [27], and modulates macrophage polarization by shifting M2 macrophages to the M1 phenotype [28]. Notably, caerin 1.1/1.9 also enhance the efficacy of anti-PD-1 therapy and vaccine-based immunotherapies [29]. It has also been found that caerin 1.1/1.9 upregulates CD47 on tumor cells; the combination of caerin 1.1/1.9 intratumor injection, CD47 blockade, and therapeutic vaccination (triple therapy, TT) significantly prolonged survival in B16 tumor-bearing mice [28]. Additionally, we have found that the TT can halt B16 melanoma metastatic tumors and induce a notable expansion of conventional type 1 dendritic cells (cDC1s) and CD4^+^CD8^+^ T cells [30].

This study aims to evaluate the antitumor effects of caerin 1.1 and 1.9 peptides against 4T-1 breast cancer cells in vitro and their impacts on tumor growth and pulmonary metastasis in vivo. Furthermore, we investigate the synergistic potential of caerin peptides with dendritic cell (DC) vaccines and other immunotherapeutic approaches. Although the combinations of DC vaccines with immune checkpoint inhibitors or chemotherapy have been explored, the strategy utilizing host defense peptides to simultaneously provide both tumor antigen sources and immunomodulatory functions remains unexplored. For the first time, we demonstrate that caerin peptides enhance DC vaccine efficacy by inducing tumor-specific cell death modalities (e.g., pyroptosis) and cooperatively reprogramming the immunosuppressive tumor microenvironment, offering a novel combinatorial strategy for poorly immunogenic breast cancers.

## 2. Materials and Methods

### 2.1. Animals

All animal experiments were approved by the Ethics Committee of Animal Experiments at the First Affiliated Hospital of Guangdong Pharmaceutical University (approval No.: GYFYG2R202326) and conducted in accordance with ethical standards. Female BALB/c mice (6–8 weeks old) were purchased from the Guangdong Province Animal Resource Center (Guangzhou, China). The mice were housed under specific pathogen-free (SPF) conditions (temperature: 22 °C, humidity: 60%) with a 12 h light/dark cycle and provided sterile food and water ad libitum. Tumor growth was carefully monitored, and the mice were euthanized when tumor diameters exceeded 15 mm, as per ethical guidelines.

### 2.2. Cell Culture

The 4T-1 murine breast cancer cell line was obtained from Procell Life Science & Technology Co., Ltd. (Wuhan, China). The cells were cultured in Dulbecco’s Modified Eagle Medium (DMEM; Gibco, Waltham, MA, USA) supplemented with 10% fetal bovine serum (FBS; Gibco) and 100 mM penicillin/streptomycin (Gibco). The cells were maintained in a humidified incubator at 37 °C with 5% CO_2_.

### 2.3. Peptide Synthesis

Caerin 1.1 (F1: GLLSVLGSVAKHVLPHVVPVIAEHL-NH_2_) and caerin 1.9 (F3: GLFGVLGSIAKHVLPHVVPVIAEKL-NH_2_), along with a control peptide P3 (GTELPSPPSVWFEAEFK-OH), were synthesized by Mimotopes Proprietary Limited (Wuxi, China). All peptides had a purity > 99% and endotoxin levels < 0.44 EU/mL.

### 2.4. MTT Assay

4T-1 cells were seeded at a density of 1 × 10^4^ cells per well in 96-well plates and cultured for 24 h. Peptides F1/F3 were added at concentrations ranging from 0 to 20 μg/mL and incubated for 18 h. MTT reagent (10 μL per well; Beyotime) was then added and incubated for 4 h. Afterward, 50 μL of dimethyl sulfoxide (DMSO; Sigma-Aldrich, St. Louis, MO, USA) was added to each well. Absorbance at 540 nm was measured using a microplate reader.

### 2.5. Apoptosis Detection

Apoptosis was assessed using an Annexin V-FITC/PI Apoptosis Detection Kit (Beyotime, Shanghai, China). 4T-1 cells (5 × 10^5^ cells/well) were seeded in 6-well plates and incubated for 24 h. After treatment with 10 μg/mL of F1/F3 for 18 h, the cells were harvested using EDTA-free trypsin (Gibco), stained with 5 μL Annexin V-FITC and 10 μL propidium iodide (PI), and incubated in the dark for 10–20 min at room temperature. Flow cytometry was performed immediately after staining.

### 2.6. Cell Morphology Analysis

4T-1 cells (1 × 10^5^ cells/well) were seeded in 12-well plates and cultured for 24 h. After washing with PBS, the cells were treated with 10 μg/mL of F1/F3. Morphological changes were observed under a fluorescence microscope in bright-field mode.

### 2.7. Tumor Model

Subcutaneous tumor model: 4T-1 cells (5 × 10^5^/200 μL) were injected subcutaneously into the lateral flanks of BALB/c mice. Tumor volumes were measured every two days using a caliper and calculated as follows: volume = length × (width^2^/2). The mice were euthanized when the tumor diameters exceeded 15 mm.

Lung metastasis model: 4T-1 cells (5 × 10^5^/200 μL) were injected into the fourth mammary fat pad of BALB/c mice. Tumor growth was monitored, and after 30 days, the mice were euthanized, and the lungs were harvested, stained, and analyzed for metastatic nodules.

### 2.8. Intratumoral Administration

Three days after tumor cell inoculation (tumor diameters: 3–5 mm), the mice received daily intratumoral injections of 30 μg F1/F3 peptides in PBS for seven consecutive days.

### 2.9. Analysis of Lung Metastasis

To assess the anti-metastatic effects, BALB/c mice with mammary tumors (tumor diameters: 3–5 mm) received daily intratumoral injections of 30 μg F1/F3 peptides in PBS for seven days. On day 30, the lungs were excised, stained with India ink, and fixed in Fekete’s solution (70 mL absolute ethanol, 5 mL glacial acetic acid, 10 mL formalin, 30 mL pure water). Metastatic nodules were counted under a microscope.

### 2.10. Tumor Antigen Preparation

TAA_1_ (total tumor antigens): 4T-1 cells (1 × 10^7^/mL) were subjected to five freeze–thaw cycles at −80 °C and 37 °C. Cell debris was removed by centrifugation at 3000 rpm for 10 min, and the supernatant was collected and stored at −80 °C.

TAA_2_ (peptide-induced apoptotic tumor antigens): 4T-1 cells were treated with 10 μg/mL of F1/F3 peptides for 1 h, then subjected to five freeze–thaw cycles as described above. The supernatant was collected and stored at −80 °C.

### 2.11. DC Vaccine Preparation and Administration

DC preparation: Bone marrow cells were isolated from BALB/c mice and cultured in RPMI-1640 medium containing 20 ng/mL IL-4 (Novoprotein, Guangzhou, China) and GM-CSF (Novoprotein) for seven days. On day 7, the DC cells were purified with OptiPrep reagent (Sigma, Tokyo, Japan), diluent (0.88% (*w*/*v*) sodium chloride (Macklin, Shanghai, China), 1 mM EDTA (Macklin), 0.5% bovine serum protein (BSA) (Macklin), 10 mM Tricine-NaOH (Macklin), and a cell tissue separation solution (2.3 volumes of OptiPrep + 9.7 volumes of diluent), co-cultured with tumor antigens (TAA_1_ or TAA_2_) at a 1:3 ratio, and stimulated with TNFα (10 ng/mL; Novoprotein), IL-1β (10 ng/mL; Novoprotein), IL-6 (100 ng/mL; Novoprotein), and PGE2 (0.2 mg/mL; Macklin).

Vaccine administration: BALB/c mice with subcutaneous 4T-1 tumors (3–5 mm) received intratumoral injections of DC vaccines on days 3 and 9.

### 2.12. Flow Cytometry

Detection was performed using flow cytometry (BD FACSaria II). The antibodies used for flow cytometry are listed in Table 1.

### 2.13. Tumor and Lymph Node Microenvironment Analysis by Flow Cytometry

Tumor and lymph node tissues were harvested from mice, processed into single-cell suspensions, stained with appropriate flow cytometry antibodies, and analyzed using flow cytometry.

### 2.14. IFN-γ Detection in the Spleen and Lymph Nodes

Anti-CD3e (final concentration: 2 μg/mL) was pre-coated onto 24-well plates by adding 1 mL PBS containing anti-CD3e per well, followed by incubation at 37 °C for 2 h. Single-cell suspensions were prepared from spleens and lymph nodes, and 1 × 10^6^ cells/mL were seeded into each well. The Cell Stimulation Cocktail (2 μL/well) was added, and the cells were cultured at 37 °C for 5–18 h. After incubation, the cells were collected, washed with Wash Buffer, and stained with surface antibodies. Afterward, the cells were fixed and permeabilized with 250 μL of a Fixation/Permeabilization solution (eBioscience) for 30–60 min, followed by washing with permeabilization buffer three times. Intracellular staining was performed with anti-IFN-γ antibodies for 15–30 min (Using Rat IgG1κ as the isotype control antibody for IFNγ detection), and the cells were analyzed by flow cytometry.

### 2.15. Magnetic Bead Sorting of T Cells

Spleens were harvested from mice, and the cells were purified using sample density gradient centrifugation (TBDscience, Tianjin, China). The purified cells were subjected to positive selection using CD4/CD8 magnetic beads (Miltenyi Biotec, Singapore) to obtain the desired T-cell populations.

### 2.16. Co-Culture Experiments

#### 2.16.1. DC Vaccines (DCV) and T Cell Co-Culture

BMDC-derived DCs loaded with TAA_1_ antigens (DCV_1_) were co-cultured with T cells isolated from the spleen using CD4/CD8 magnetic beads at a 1:5 ratio. After 3 days of co-culture, the cells were collected and analyzed by flow cytometry.

#### 2.16.2. DCV, T Cells, and Tumor Cell Co-Culture

DCV_1_ (TAA_1_) and DCV_2_ (TAA_2_) were co-cultured with T cells (isolated using CD4/CD8 magnetic beads) and 4T-1 or TC-1 tumor cells at a ratio of 2:10:1. Tumor cells and T cells were pre-stained with CFSE and eFluor450 dyes, respectively. After 3 days of co-culture, the cells were collected for flow cytometry analysis.

### 2.17. ELISA

Supernatants from the co-culture experiments of DCV, T cells, and tumor cells were collected. The levels of immune-related cytokines, including TNFα, IL-12, IL-6, and IL-10, were measured using ELISA kits (BioLegend, San Diego, CA, USA) following the manufacturer’s instructions.

## 3. Results

### 3.1. F1/F3 Induces Programmed Cell Death in 4T-1 Cells In Vitro

4T-1 cells were treated with varying concentrations of F1, F3, and F1/F3 for 18 h. A significant inhibitory effect on 4T-1 cell proliferation was observed using a MTT assay. The half-maximal inhibitory concentrations (IC50) of F1, F3, and F1/F3 were 9.227 ± 0.375 μg/mL, 14.02 ± 1.89 μg/mL, and 7.358 ± 0.115 μg/mL, respectively. Notably, the combination of F1 and F3 exhibited a stronger inhibitory effect compared to the F1 or F3 treatments (Figure 1A–C). Real-time microscopic observation revealed that treatment with 10 μg/mL of F1/F3 induced pronounced membrane morphological changes in 4T-1 cells, the cell volume increased, and the cytoplasm expanded, while no morphological alterations were observed in the untreated (UN) group or the P3 group. P3 is a peptide with no cytotoxicity [31] (Figure 1D). Annexin V/PI staining further confirmed the pro-apoptotic effects of F1/F3 on 4T-1 cells. Treatment with 10 μg/mL of F1/F3 significantly induced apoptosis, with an apoptosis rate of 32.43%, compared to 13.29% in the untreated group and 12.48% in the P3 group (Figure 1E). This finding is consistent with our research group’s previous studies demonstrating that caerin peptides possess the ability to induce both apoptosis and pyroptosis in cancer cells [32]. Collectively, these findings demonstrate that F1/F3 effectively inhibits 4T-1 cell proliferation and induces apoptosis in vitro.

### 3.2. F1/F3 Inhibits 4T-1 Tumor Growth and Lung Metastasis in Mice

To investigate the in vivo effects of F1/F3 on tumor growth and metastasis, a 4T-1 tumor murine model was established by subcutaneously inoculating 4T-1 cells on the flank or the fourth mammary fat pad of mice (Figure 2A). The tumors were treated with intratumoral injections of F1/F3 for 7 days. As shown in Figure 2B, which depicts tumor implantation via the flank, treatment with F1/F3 significantly suppressed tumor growth compared to the PBS and P3 control groups. Furthermore, F1/F3 treatment significantly prolonged the survival of 4T-1 tumor-bearing mice, with a median survival of 37 days, compared to 27 days for both the PBS and P3 groups (Figure 2C). On day 18 post inoculation, the tumors were excised and weighed. The tumors in the F1/F3 group were significantly smaller than those in the PBS and P3 groups, with an average tumor weight of 0.0471 ± 0.0240 g, compared to 0.1221 ± 0.0586 g and 0.1256 ± 0.0689 g in the PBS and P3 groups, respectively (Figure 2D and Appendix A). Distant metastasis, including to the lungs, bones, liver, and brain is often observed in breast cancer [33]. To assess the potential of F1/F3 to inhibit metastasis, we evaluated pulmonary metastasis in the 4T-1 model, where 4T-1 cells were inoculated to the fourth mammary fat pad. As shown in Figure 2E,F, the F1/F3 treatment significantly prolonged survival compared to the UN and P3 groups, with a median survival of 36 days for the F1/F3 group, compared to 28 and 30 days for the UN and P3 groups, respectively. Lung metastasis was also significantly reduced in the F1/F3-treated mice. The average number of lung metastases in the F1/F3 group was two, compared to 7 and 11 in the UN and P3 groups, respectively (Figure 2G and Appendix A). The specific numbers of lung tumor nodules in each group are detailed in Appendix A. These results suggest that F1/F3 effectively inhibits tumor growth, prolongs survival, and reduces lung metastasis in 4T-1 tumor-bearing mice.

### 3.3. Intratumoral Injection of F1/F3 Promotes cDC1 and CD4^+^ T Cells Infiltrating the Tumor

Based on the observation that F1/F3 suppressed 4T-1 tumor progression and our previous published papers [28], we further investigated the changes in the microenvironment across relevant organs by flow cytometry. The results showed that, in tumor tissues, the proportions of total T cells (28.88% vs. 2.93% vs. 5.16%) and the CD4^+^ T cell (89.23% vs. 36.97% vs. 34.33%) subset were significantly higher in the F1/F3-treated mice group compared with both the PBS- and P3-treated mice (Figure 3A,B and Appendix A). By contrast, the proportions of CD8^+^ T cells (6.94% vs. 14.43% vs. 11.25%) were significantly lower in the F1/F3 group (Figure 3C and Appendix A). Interestingly, the macrophage population was notably elevated in the F1/F3 group compared to the PBS and P3 groups (3.38%, 3.01%, and 2.52%, respectively), yet the proportion of pro-inflammatory macrophages (M1) (CD45^+^CD11b^+^F4/80^+^Ly6C^+^) decreased significantly (8.24% vs. 14.67% vs. 18.68%), whereas non-inflammatory macrophages (M2) (CD45^+^CD11b^+^F4/80^+^Ly6C^−^) increased (90.45% vs. 83.70% vs. 79.28%) (Figure 4A). Notably, although the overall proportion of dendritic cells (DCs) in the tumor microenvironment appeared lower following F1/F3 treatment (0.67% vs. 1.78% vs. 2.67%), the F1/F3 group exhibited a significantly higher percentage of cDC1 (18.55% vs. 7.77% vs. 6.91%) and a lower percentage of cDC2 (52.25% vs. 84.75% vs. 84.38%) (Figure 4B). In addition, compared with the PBS and P3 groups, F1/F3 treatment markedly reduced the proportion of migratory cDC1 (20.88% vs. 81.90% vs. 70.90%) and increased tissue-resident cDC1 (55.38% vs. 0.35% vs. 0.67%). A similar pattern was observed for migratory and tissue-resident cDC2 subpopulations (Figure 4C).

### 3.4. The Cell Population Changes in the Lymph Nodes

Compared with the PBS group, the F1/F3 group showed increased proportions of CD45.2^+^ cells and T cells (99.55% vs. 98.1%, 32.43% vs. 27.93%, respectively), and a similar trend was observed in the non-draining lymph nodes (Figure 5A,B and Figure 6A,B). Regarding T cell subsets, the proportion of CD4^+^ T cells was significantly higher in the F1/F3 group than in the PBS group (59.15% vs. 49.15%), whereas the proportion of CD8^+^ T cells was significantly lower (31.25% vs. 39.13%), and this pattern was also noted in the non-draining lymph nodes (Figure 5C and Figure 6C). Meanwhile, among dendritic cell (DC) subsets, cDC1 expression was significantly elevated in the F1/F3 group compared with the PBS group (69.70% vs. 64.08%), whereas cDC2 expression was markedly reduced (29.28% vs. 34.93%), and similar trends were observed in the non-draining lymph nodes (Figure 5D and Figure 6D).

Furthermore, the level of PD-1 expression on T cells was significantly higher in the F1/F3 group (5.16%) than in the PBS group (4.69%) (Figure 5E and Appendix A). In contrast, the level of PD-L1 expression on DCs was significantly lower in the F1/F3 group (33.28%) than in the PBS group (40.58%) (Figure 5F) in draining lymph nodes, and this was also true in the non-draining lymph nodes (Figure 6E,F). Analyzing PD-L1 expression across various cell subsets revealed that in both draining and non-draining lymph nodes, the PD-L1 levels on cDC1 were significantly lower in the F1/F3 group than in the PBS group (Figure 5G and Figure 6G). Interestingly, cDC2 showed the opposite trend: PD-L1 expression increased in the draining lymph nodes of the F1/F3 group but decreased in the non-draining lymph nodes (Figure 5G and Figure 6G). Compared with the PBS group, PD-1 expression on CD4^+^ and CD8^+^ T cells in the F1/F3 group did not differ significantly in the draining lymph nodes but was significantly upregulated in the non-draining lymph nodes (Figure 5G, Figure 6G and Appendix A). Taken together, these data indicate that F1/F3 effectively remodels the murine immune microenvironment by increasing the numbers of cDC1 and CD4^+^ T cells in the lymph nodes.

To further validate these observations, we established a B16 tumor-bearing mouse model was used. In the draining lymph nodes, the numbers of T cells expression levels were significantly higher in the F1/F3 group than in the P3 group (6.35% vs. 4.38%). Although the CD4^+^ T cells displayed an upward trend, it did not reach statistical significance; however, the proportion of CD8^+^ T cells was significantly higher in the F1/F3 group (26.80%) compared with the P3 group (12.45%). However, the subsets of T cells and their PD-1 expression exhibited opposite patterns compared to those observed in the 4T-1 model (Appendix A). Additionally, in the draining lymph nodes, cDC1 expression was markedly higher in the F1/F3 group than in the P3 group (83.95% vs. 74.18%), whereas cDC2 expression was notably lower (14.08% vs. 22.68%). Both DC subsets also showed significantly reduced PD-L1 expression in the F1/F3 group (30.75% vs. 45.30%, 68.50% vs. 90.95%). A similar pattern was observed in the non-draining lymph nodes, although it did not reach significance (Appendix A). The results here are similar to those observed in the 4T-1 model.

### 3.5. PD-1 Blockade Does Not Increase the Efficacy of F1/F3

Given that PD-1 expression in the lymph node microenvironment was markedly elevated following F1/F3 treatment, we administered F1/F3 combined with anti-PD-1 via intratumoral injection. Although both F1/F3 plus anti-PD-1 and F1/F3 alone were able to delay tumor progression, there was no statistically significant difference between the two regimens; the median survival times of mice in the F1/F3 plus anti-PD-1 group and the F1/F3-alone group were 38 and 40 days, respectively (Appendix A). Moreover, F1/F3 combined with anti-PD-1 did not more effectively suppress tumor metastasis to the lungs than F1/F3 alone, with mean metastatic nodules of 5 versus 2, respectively (Appendix A).

### 3.6. DC Vaccines Effectively Suppress 4T-1 Tumor Growth

Previously, we demonstrated that F1/F3 enhances the efficacy of a therapeutic vaccine in a HPV16+ TC-1 model [28,29]; therefore, we hypothesize that F1/F3 increases the efficacy of therapeutic vaccine in the 4T-1 tumor model, 4T-1 tumors lack unique, tumor-specific antigens, and most existing therapeutic vaccines against 4T-1 tumor are dendritic-cell-based vaccines [34,35]. DCs pulsed with freeze/thawed TC-1 cells were co-cultured it with T cells in vitro. Compared with DCs without antigen loading and T cells alone, the TAA-loaded DCs effectively promoted T cell proliferation and enhanced IFN-γ secretion (Appendix A).

To further assess the therapeutic efficacy of the whole tumor-cell-based DC vaccine in 4T-1 tumor-bearing mice, A DC vaccine loaded with 4T-1 cells was administered it to 4T-1 tumor-bearing mice (Figure 7A). The results showed that the DC vaccine effectively inhibited 4T-1 tumor growth and extended mouse survival time. Specifically, the median survival time of the DC vaccine group was 41 days, compared to 30 days in the untreated group (UN), although this did not differ significantly from the median survival of 39 days in the F1/F3-treated group (Figure 7B,C).

### 3.7. DC Vaccine Improves T Cell Response

Next, we performed flow cytometric analyses of the immune microenvironment, including the draining lymph nodes, non-draining lymph nodes, and spleen. In the draining lymph nodes, IFN-γ expression was lower in the DCV group (9.91%) than in the UN group (25.30%), whereas in the non-draining lymph nodes, IFN-γ expression was significantly higher in the DCV group (11.30% vs. 2.10%) (Figure 7D and Appendix A). Further analysis revealed that, in the draining lymph nodes, the DCV group showed markedly higher proportions of CD45.2^+^ immune cells, T cells, and CD4^+^ T cells compared with the UN group (99.70% vs. 98.65%, 50.35% vs. 35.78%, 61.38% vs. 42.55%, respectively), along with a significantly lower proportion of CD8^+^ T cells (33.88% vs. 48.50%) (Figure 7E). A similar pattern was observed in the non-draining lymph nodes (Appendix A). Examining the various T cell subsets, the DCV group had significantly more effector T cells (CD45.2^+^CD3e^+^CD44^high^CD62L^−^) in the draining lymph nodes than the UN group (68.73% vs. 55.08%), but significantly fewer in the non-draining lymph nodes (70.78.1% vs. 83.15%). Moreover, in the draining lymph nodes, the proportion of effector CD4^+^ T cells (CD45.2^+^CD3e^+^CD44^high^CD62L^−^CD4^+^) in the DCV group (83.20%) exceeded that in the UN group (63.28%), while effector CD8^+^ T cells (CD45.2^+^CD3e^+^CD44highCD62L^−^CD8^+^) trended downward (Figure 8A). In the non-draining lymph nodes, effector CD4^+^ T cells remained relatively unchanged, whereas effector CD8^+^ T cells increased substantially (16.43% vs. 11.13%) (Appendix A).

A closer examination revealed that in the draining lymph nodes, the DCV group had significantly higher proportions of naive T cells (CD45.2^+^CD3e^+^CD62LhighCD44^−^) and naive CD4^+^ T cells (CD45.2^+^CD3e^+^CD62LhighCD44^−^CD4^+^) than the UN group (93.68% vs. 89.80%, 49.00% vs. 5.13%), while naive CD8^+^ T cells (CD45.2^+^CD3e^+^CD62LhighCD44^−^CD8^+^) were markedly lower (47.23% vs. 88.98%) (Figure 8B). The same trend was observed in the non-draining lymph nodes (Appendix A). Regarding effector memory T cells and their subsets, the DCV group showed a pronounced increase in effector memory T cells (CD45.2^+^CD3e^+^CD44highCD62L^−^CCR7^−^) within the draining lymph nodes (68.75% vs. 55.08%), but a notable decrease in the non-draining lymph nodes (71.28% vs. 83.68%). Likewise, effector memory CD4^+^ T cells (CD45.2^+^CD3e^+^CD44highCD62L^−^CCR7^−^CD4^+^) rose significantly in the draining lymph nodes (82.75% vs. 59.85%), while effector memory CD8^+^ T cells (CD45.2^+^CD3e^+^CD44highCD62L^−^CCR7^−^CD8^+^) showed a downward trend; in the non-draining lymph nodes, effector memory CD4^+^ T cells slightly decreased, whereas effector memory CD8^+^ T cells significantly increased (16.43% vs. 11.23%) (Figure 8C and Appendix A).

Finally, we performed the same analyses on the spleen and found no statistically significant differences between the DCV group and the UN group in IFN-γ expression, CD45.2^+^ cells, or T cell subsets (Appendix A). Taken together, these findings suggest that intra-tumor injection DC vaccine loaded with 4T-1 whole-tumor antigens augments the immune response in vivo, especially in draining LNs, and exerts a notable inhibitory effect on 4T-1 tumor growth.

### 3.8. DC Vaccine Pulsed with F1/F3-Treated 4T-1 Cells Outperforms 4T-1-Loaded DC Vaccine

Previous studies have demonstrated that F1/F3 can promote the secretion of pro-inflammatory cytokines by tumor cells [29]. We hypothesized that DCs pulsed with F1/F3 stimulated 4T-1 cells might more effectively inhibit the growth of 4T-1 tumors. Two types of DC vaccines, one loaded with F1/F3-treated 4T-1 cells (DCV_2_) and the other loaded with 4T-1 cells (DCV_1_), were compared for their ability to inhibit 4T-1 tumor growth. Both DCV_1_ and DCV_2_ significantly inhibited tumor growth and prolonged the survival of 4T1 tumor-bearing mice. However, there was no significant difference between DCV_1_ and DCV_2_, nor was there a statistically significant difference compared to the F1/F3-alone group (Figure 9B,C). Specifically, the median survival time was 49 days for the DCV_1_ group, 48 days for the DCV_2_ group, and 43 days for the F1/F3 group, while the UN group had a median survival of only 31 days. Next, we investigated whether combining DC vaccines with F1/F3 could yield better therapeutic outcomes in the 4T-1 tumor model (Figure 9A). The results revealed that both the DCV_1_ and DCV_2_, when combined with F1/F3, exhibited superior tumor inhibition compared to the F1/F3-alone group. Notably, the difference between DCV_2_ + F1/F3 and F1/F3 alone reached statistical significance, while DCV_1_ + F1/F3 also showed a trend toward better efficacy but did not reach statistical significance (Figure 9D).

### 3.9. DCV_2_ Stimulates Stronger T Cell Responses Compared with DCV_1_

To investigate the mechanisms underlying the superior tumor inhibitory effects of DCV_2_ compared to DCV_1_, we co-cultured DCV_1_ or DCV_2_ with T cells and either TC-1 or 4T-1 tumor cells in vitro. The experimental groups (DCV_1_ + T + tumor cells, DCV_2_ + T + tumor cells) both demonstrated cytotoxic effects against 4T-1 cells, with the DCV_2_ group (94.2%) showing a marginal increase in activity compared to the DCV_1_ group (93.8%). Additionally, DCV_2_ also significantly promoted T cell proliferation (2.61% vs. 2.3%) compared to DCV_1_ (Figure 10A,B). A similar trend was observed in the TC-1 model. Compared to the control groups (Tumor, T + Tumor), the experimental groups (DCV_1_ + T + Tumor, DCV_2_ + T + Tumor) exhibited significant cytotoxic effects against TC-1 cells (97.4% vs. 98.8% vs. 98.7%, 97.2% vs. 98.8% vs. 98.7%) and significantly promoted T cell proliferation (1.16% vs. 0.74%, 1.24% vs. 0.74%). Among them, DCV_2_ + T + Tumor (97.2%) demonstrated stronger cytotoxic activity against 4T-1 tumor cells compared to DCV_1_ + T + Tumor (97.4%). Additionally, in terms of T cell proliferation, DCV_2_ + T + Tumor (1.24%) also showed superior efficacy compared to DCV_1_ + T + Tumor (1.16%) (Figure 10A,B).

ELISA analysis of the culture supernatants showed that, in the 4T-1 model, both experimental groups significantly enhanced the secretion of immune-related inflammatory factors such as TNF-α and IL-12 by T cells. However, the secretion levels of these factors in the DCV_2_ + T + Tumor group were lower than those in the DCV_1_ + T + Tumor group (Figure 10C,D). Interestingly, the 4T-1 model also showed increased IL-10 production in both experimental groups, with DCV_2_ + T + Tumor secreting significantly less IL-10 than DCV_1_ + T + Tumor (Figure 10E). A similar pattern was observed in the TC-1 system. The experimental groups significantly enhanced the secretion of immune-related inflammatory factors such as TNF-α and IL-12 by T cells. However, the secretion levels in the DCV_2_ + T + Tumor group were lower than those in the DCV_1_ + T + Tumor group. Similarly, the experimental groups also produced more IL-10, with the DCV_2_ + T + Tumor group secreting significantly less IL-10 compared to the DCV_1_ + T + Tumor group (Figure 10C–E).

Further analysis revealed that, compared to the control groups, both experimental groups significantly upregulated PD-L1 expression on 4T-1 cells (0.085% vs. 0.13% vs. 1.95% for DCV1 and DCV2, respectively) with a similar trend in TC-1 cells (Appendix A).

## 4. Discussion

In this study, we first investigated the effects of F1/F3 on the proliferation of the murine breast cancer cell line 4T-1 in vitro. Our results showed that F1/F3 significantly inhibited 4T-1 cell growth (Figure 1A–C) and induced 4T1 apoptosis (Figure 1E). More intriguingly, F1/F3 induced pyroptosis-like morphological changes in 4T-1 cells (Figure 1D). Pyroptosis, defined as a lytic and inflammatory form of programmed cell death, is characterized by cell swelling, membrane perforation, and the release of intracellular contents [36]. As expected, F1/F3 significantly suppressed 4T-1 tumor growth and lung metastasis (Figure 2).

Flow cytometric analysis of the tumor microenvironment (TME) revealed that F1/F3 increased T cell infiltration, particularly by boosting CD4^+^ T cell numbers while reducing CD8^+^ T cells (Figure 3A–C). One possible explanation is that F1/F3 recruits T cells into the tumor and leverages CD4^+^ T cells to support and maintain CD8^+^ T-cell cytotoxicity [37,38]. Nevertheless, CD8^+^ T cells generally have a limited functional lifespan [39]; persistent antigenic stimulation by tumor cells can drive exhaustion and a subsequent decrease in CD8^+^ T-cell numbers [40]. Moreover, F1/F3 treatment augmented the macrophage population while decreasing pro-inflammatory M1 cells and increasing M2 cells (Figure 4A)—an observation that may be attributed to tumor-derived chemokines (e.g., IL-10, CCL2/3/4/5/7/8, CXCL12, VEGF, PDGF, CSF1) that recruit monocytes or M0 macrophages into the TME and induce M2 polarization [41].

Tumor Necrosis Factor Receptor-Associated Factor 2 (TRAF2) promotes the polarization of M2 macrophages, and this chemotaxis is achieved through an autophagy-dependent pathway [42]. In breast cancer, the overexpression of TRAF2 enhances the malignant migration of tumor cells and the formation of osteoclasts, thereby facilitating the osteolytic metastasis of breast cancer [42]. This may also be a contributing factor to the increase in M2 macrophages. In previous studies, F1/F3 promoted the expression of M1 macrophages and suppressed M2 macrophages in the tumor microenvironment of TC-1 and B16 animal models, whereas it exhibited the opposite effect in the 4T-1 model. This discrepancy may be attributed to the high activity of CD8+ T cells in TC-1 and B16 models [28,31], which release higher levels of IFN-γ to directly inhibit M2 polarization.

In contrast, the 4T-1 model showed a significant reduction in CD8+ T cell infiltration and diminished IFN-γ secretion, thereby potentially leading to enhanced M2 polarization. Additionally, TC-1 and B16 tumors exhibit stronger immunogenicity compared to 4T-1 tumors, which may enhance T cell responses and subsequent IFN-γ production to activate M1 macrophages [43,44,45]. Furthermore, the baseline M1/M2 ratio in C57BL/6 mice (used for TC-1 and B16 models) is inherently higher, while BALB/c mice (used for 4T-1 models) are naturally predisposed to M2 polarization [46,47].

On the other hand, F1/F3 significantly promoted the upregulation of cDC1 while markedly downregulating the expression of cDC2 within the tumor (Figure 4B). It has been reported that cDC1 is primarily responsible for cross-presenting tumor antigens to naïve CD8^+^ T cells, generating cytotoxic T lymphocytes (CTLs) [48], whereas cDC2 can present exogenous antigens via MHCII to CD4^+^ T cells [49,50]. Hence, cDC2 depletion in the tumor might reflect their migration to lymph nodes to present antigen to CD4^+^ T cells. Previous work suggests that migratory DCs (mDCs) shuttle antigens to lymph nodes via vesicles, handing them over to resident DCs (rDCs) for subsequent activation of CD8^+^/CD4^+^ T cells [50]. As the tumor grows, resident DCs continually accrue antigens and trigger antitumor CD8^+^ T-cell responses [51], which may explain why mDCs decrease while rDCs remain capable of activating T cells (Figure 4C). Our recent study has revealed that F1/F3 inhibits metastatic progression in B16 melanoma by expanding cDC1 and reprogramming TAMs. Mechanistically, these peptides enhance crosstalk between cDC1 and CD8^+^ T cells, thereby establishing an antitumor immune network [30]. These findings further demonstrate that F1/F3 significantly promotes the expression of cDC1. Why CD4^+^T cells and cDC1 are specifically attracted to 4T-1 tumor remains elusive, and chemokine level changes following the treatment of F1/F3 will be the focus of our upcoming study.

The lymphatic system is critical for maintaining homeostasis and immunosurveillance, making it highly relevant to cancer treatments in the immunotherapy era [52]. By examining both tumor-draining lymph nodes (TDLNs) and non-draining lymph nodes (non-TDLNs), we found that F1/F3 also markedly increased CD45.2^+^ cells and T cells (Figure 5A,B and Figure 6A,B). This observation may imply that F1/F3 not only modulates immune cells locally within the TME but also systematically reprograms lymphoid organs to strengthen overall antitumor immunity. CD4^+^ T cells, in addition to providing help, can secrete cytokines and deliver costimulatory signals to sustain and amplify CD8^+^ T-cell-mediated tumor killing [53,54]. However, prolonged tumor stimulation can drive CD8^+^ T-cell exhaustion, leading to functional impairment and numerical reduction [55,56]. Thus, F1/F3-driven enrichment of helper T cells and progressive CD8^+^ T-cell exhaustion may collectively explain why CD4^+^ T cells increase while CD8^+^ T cells decrease (Figure 5C,D and Figure 6C,D). The parallel patterns in both TDLNs and non-TDLNs suggest that F1/F3 exerts a systemic immunoregulatory effect rather than being confined to the tumor site, an important factor in preventing metastasis and bolstering immunosurveillance. Additionally, F1/F3 upregulated PD-1 but downregulated PD-L1 on DCs (Figure 5E,F and Figure 6E,F). Previous research reveals that PD-1^+^ DCs impair CD8^+^ T-cell function and infiltration while diminishing antigen presentation and MHCI expression [57,58], thereby enabling immune escape. Moreover, PD-L1 loss on DCs can significantly restrain tumor growth and enhance CD8^+^ T-cell antitumor responses [59]. In TDLNs, F1/F3 decreased PD-L1 on cDC1 yet increased it on cDC2, whereas in non-TDLNs, PD-L1 was reduced on both subsets (Figure 5G and Figure 6G). Other studies have shown that PD-L1 expression on DCs from peripheral blood and tumor tissues can be ≥20 times higher than B7-1, and PD-L1 can bind B7.1 in cis, limiting B7.1’s interaction with T-cell CD28 and weakening T-cell activation [60]. Although F1/F3 combined with anti-PD-1 further hindered 4T-1 growth and metastasis, its effect was not superior to that of F1/F3 alone (Appendix A), potentially due to tumor-mediated epigenetic upregulation of PD-L1 for instance, removing inhibitory DNA/histone modifications at the CD274 locus in breast cancer stem cells [61]. Interestingly, recent findings show that DCs can enhance anti-PD-1 efficacy [62,63], suggesting avenues for future combination immunotherapy.

DCs play pivotal roles in bridging innate and adaptive immunity and are thus considered prime candidates for cancer vaccine development [64]. Here, we constructed DC cells loaded with tumor-associated antigens (TAAs), referred to as DCV/TAA, which significantly promoted T-cell proliferation and IFN-γ production in vitro (Appendix A). In vivo, this DCV also markedly suppressed 4T-1 tumor growth (Figure 7A–C). Flow cytometric analysis showed that DCV lowered IFN-γ secretion in TDLNs but elevated it in non-TDLNs (Figure 7D and Appendix A), indicating that DC vaccination may induce a more systemic immune response. Moreover, DCV increased CD45.2^+^ cells, T cells, and CD4^+^ T cells while decreasing CD8^+^ T cells in both TDLNs and non-TDLNs (Figure 7E and Appendix A). Regarding T-cell subsets, naïve T cells and their CD4^+^ subpopulation rose significantly, whereas naïve CD8^+^ T cells declined (Figure 8B and Appendix A)—possibly because T cells are activated in lymph nodes then migrate to tumors [48], or because they gradually enter an exhausted state under continuous antigenic and inflammatory stimulation [65].

Notably, in TDLNs, effector/effector memory T cells and CD4^+^ T cells markedly increased, while effector/effector memory CD8^+^ T cells decreased (Figure 8A,C). In non-TDLNs, effector/effector memory T cells and CD4^+^ T cells trended downward, whereas effector/effector memory CD8^+^ T cells increased (Appendix A), suggesting their trafficking to the tumor site for cytolytic action. By contrast, no significant differences were observed in the spleen (Appendix A). This phenomenon may be attributed to the fact that TDLNs serve as the primary site for tumor antigen accumulation and antigen-presenting cell (e.g., dendritic cells, DCs) aggregation. DC vaccines loaded with tumor antigens preferentially migrate to TDLNs, where they prime local T cells and remodel the immunosuppressive microenvironment [66]. In contrast, activated T cells or cytokines (e.g., IFN-γ, IL-12) may disseminate to non-TDLNs via systemic circulation, thereby initiating a broader immune response [67]. However, the spleen acts as a major reservoir for regulatory T cells (Tregs) and myeloid-derived suppressor cells (MDSCs), and immunosuppressive factors such as TGF-β and IL-10 secreted by 4T-1 tumors further promote the expansion of these inhibitory cell populations in the spleen [68], ultimately counteracting the immunostimulatory effects of DC vaccines.

In vivo comparisons of DCV_1_ or DCV_2_ monotherapy with F1/F3 monotherapy revealed that neither DC vaccine surpassed the effect of F1/F3 alone (Figure 9B,C). Nevertheless, combining DCV_1_ with F1/F3 provided a modest improvement in controlling tumor growth compared to F1/F3 alone, while DCV_2_ + F1/F3 exhibited the strongest suppression, showing statistical significance over F1/F3 monotherapy (Figure 9A,D). Building on our previous findings, combination therapy with a therapeutic vaccine and anti-IL-10 antibody induced stronger antigen-specific T cell responses in TC-1 tumors compared to the vaccine alone. Further integration of F1/F3 enhanced antitumor efficacy, suggesting that IL-10 suppression plays a critical role in potentiating immune activation [31]. Based on these observations, we hypothesize that the superior efficacy of DCV2 over DCV1 in the 4T-1 model may stem from the reduced IL-10 secretion in DCV2-treated tumors, thereby alleviating immunosuppression and amplifying T-cell-mediated antitumor activity.

To investigate the mechanistic basis for DCV_2_’s advantage over DCV_1_, we performed in vitro co-culture assays with DCV_1_ or DCV_2_, T cells, and tumor cells. Both vaccines effectively killed tumor cells and promoted T-cell proliferation, yet DCV_2_ was consistently more potent (Figure 10A,B). However, although DCV_2_ exhibited a stronger T-cell stimulatory capacity than DCV_1_ in vitro, this superior effect was not reflected in in vivo tumor suppression. This discrepancy may be attributed to the strongly immunosuppressive tumor microenvironment (TME) and the inherent complexity of antigen presentation under physiological conditions [69,70]. Additionally, both DCV_1_ and DCV_2_ upregulated PD-L1 (Appendix A), leaving open the possibility of integrating anti-PD-L1 therapy. It has been noted that Ly-6A, a GPI-anchored membrane protein expressed on naïve T cells and highly upregulated upon activation, can induce apoptosis in CD4^+^ T cells by releasing cytochrome c and activating caspases 9 and 3 [71]. Our ELISA data indicated that both DCV_1_ and DCV_2_ triggered T-cell secretion of TNF-α and IL-12, though DCV_2_ released less of these cytokines (Figure 10C,D). As TNF-α activity correlates positively with Ly-6A-mediated T-cell apoptosis [71], DCV_2_’s lower TNF-α output may mitigate T-cell apoptosis and thus drive superior T-cell proliferation relative to DCV_1_. Meanwhile, IL-12 is essential for Th1 differentiation [72], functioning through STAT4 activation and IFN-γ induction [73,74]. Under tumor-mediated interference, DCV_2_ may favor Th1 polarization yet release reduced amounts of IL-12, without hindering T-cell proliferation. Further, IL-10, a suppressive cytokine that dampens CD4^+^ T-cell expansion and function [75], was significantly lower in DCV_2_-treated groups (Figure 10E), suggesting that DCV_2_ more effectively counters immunosuppression within the tumor microenvironment. Taken together, DCV_2_ demonstrated superior antitumor and immunomodulatory potential both in vitro and in vivo, providing a solid theoretical foundation for combining DCV_2_ with F1/F3 and PD-L1 blockade in future studies.

## 5. Conclusions

The findings of this study demonstrate that F1/F3 significantly inhibits the proliferation of 4T-1 tumor cells in vitro and markedly suppresses tumor growth and lung metastasis in vivo, thereby prolonging the survival of mice. To further enhance the therapeutic efficacy against 4T-1 tumors, we also investigated the application of a DC vaccine. The results revealed that the DC vaccine alone did not significantly improve the inhibition of 4T-1 tumors. However, when combined with F1/F3, the therapeutic efficacy was significantly superior to that of either F1/F3 or the DC vaccine alone. Future improvements and refinements in the combined strategy of the DC vaccine and F1/F3 may further enhance their synergistic antitumor effects, offering a more promising therapeutic approach.

## Figures and Tables

**Figure 1 vaccines-13-00577-f001:**
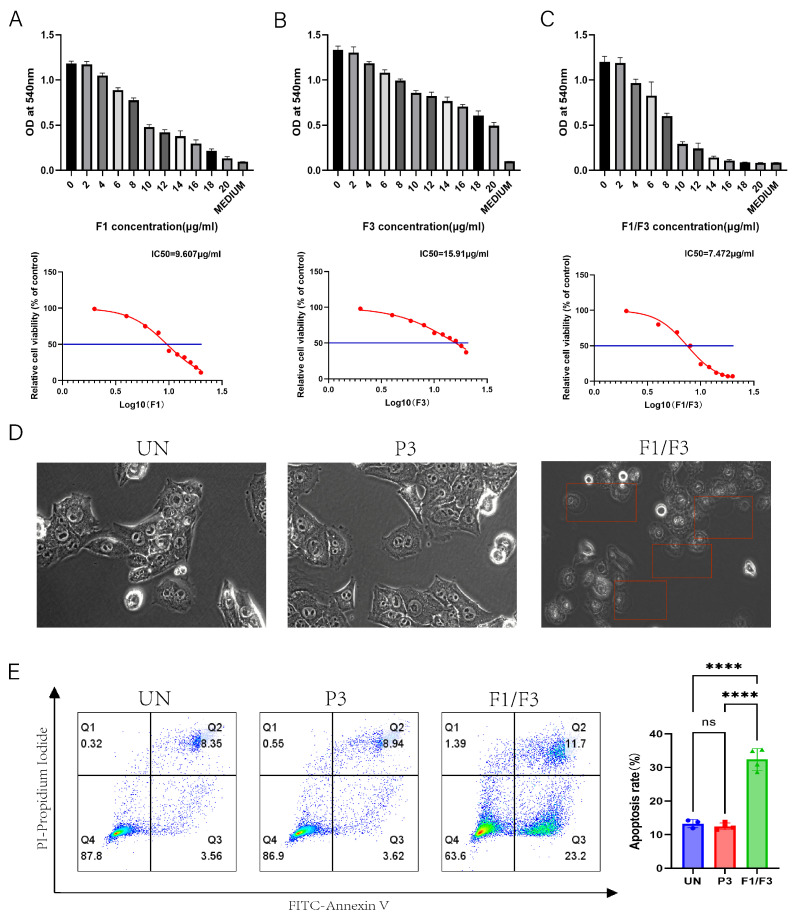
F1, F3, and F1/F3 inhibit 4T-1 cell proliferation. F1, F3, and F1/F3 inhibit the proliferation of 4T-1 cells, as assessed by the MTT assay. (**A**) The half-maximal inhibitory concentration (IC50) of F1 alone was 9.607 μg/mL; (**B**) F3 alone showed an IC50 of 15.91 μg/mL; (**C**) the combination of F1 and F3 (F1/F3) yielded an IC50 of 7.472 μg/mL; (**D**) representative microscopy images of 4T-1 cells under: no treatment (UN), P3, and F1/F3 at 10 μg/mL revealing morphological changes; Red boxes: cells with changed morphology; Microscope at 20× magnification; (**E**) flow cytometry results comparing apoptosis in 4T-1 cells among the UN, P3, and F1/F3 groups. Left: streaming scatter plot of different groups; Right: bar chart, UN group (blue), P3 group (red), F1/F3 group (green). Data in (**A**–**E**) represent an independent experiment repeated twice. The results are expressed as mean ± SD. ns, not significant; **** *p* < 0.0001. Statistical analyses were performed using one-way ANOVA for (**A**–**C**) and Student’s *t*-test for (**E**).

**Figure 2 vaccines-13-00577-f002:**
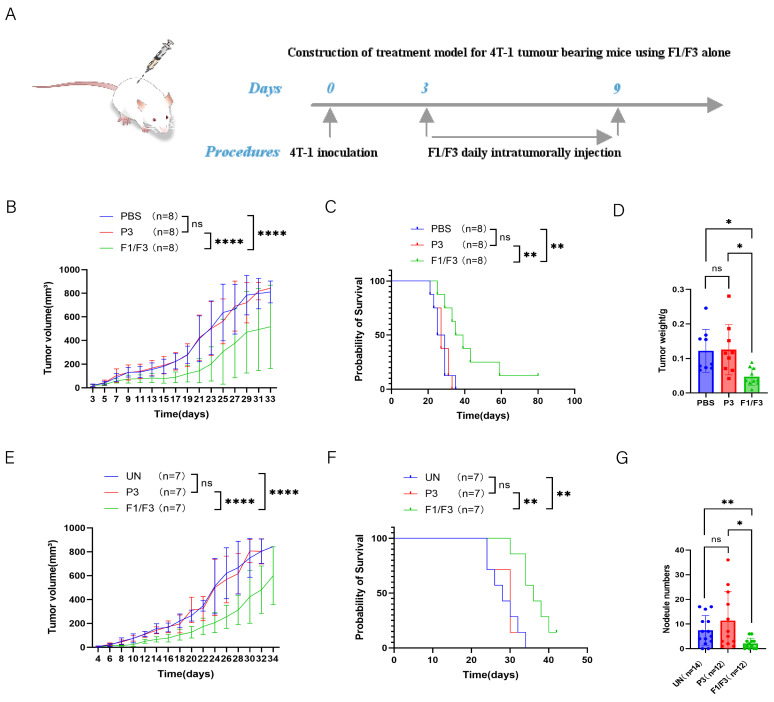
F1/F3 inhibits 4T-1 tumor growth in vivo. (**A**) 4T-1 cells (5 × 10^5^/200 μL) were injected subcutaneously into the lateral flanks of BALB/c mice, followed by local administration of PBS, P3, or F1/F3; (**B**) tumor volume and (**C**) survival curves; (**D**) tumor weights were measured on Day 18. 4T-1 cells (5 × 10^5^/200 μL) were injected into the fourth mammary fat pad of BALB/c mice, PBS, P3, or F1/F3 was administered intratumorally; (**E**) tumor growth and (**F**) survival rates in this model; (**G**) on day 30, lungs were isolated and stained with 15% India ink, and pulmonary nodules were counted (n: number of experimental mice). Each group has 3–8 mice. Data in (**A**–**F**) represent a single independent experiment, whereas (**G**) pools data from two independent experiments. The results are shown as mean ± SD. ns, not significant; * *p* < 0.05; ** *p* < 0.01; **** *p* < 0.0001. Statistical analyses were carried out using two-way ANOVA for (**B**,**E**), Kaplan–Meier survival analysis for (**C**,**F**), one-way ANOVA for (**D**), and Student’s *t*-test for (**G**).

**Figure 3 vaccines-13-00577-f003:**
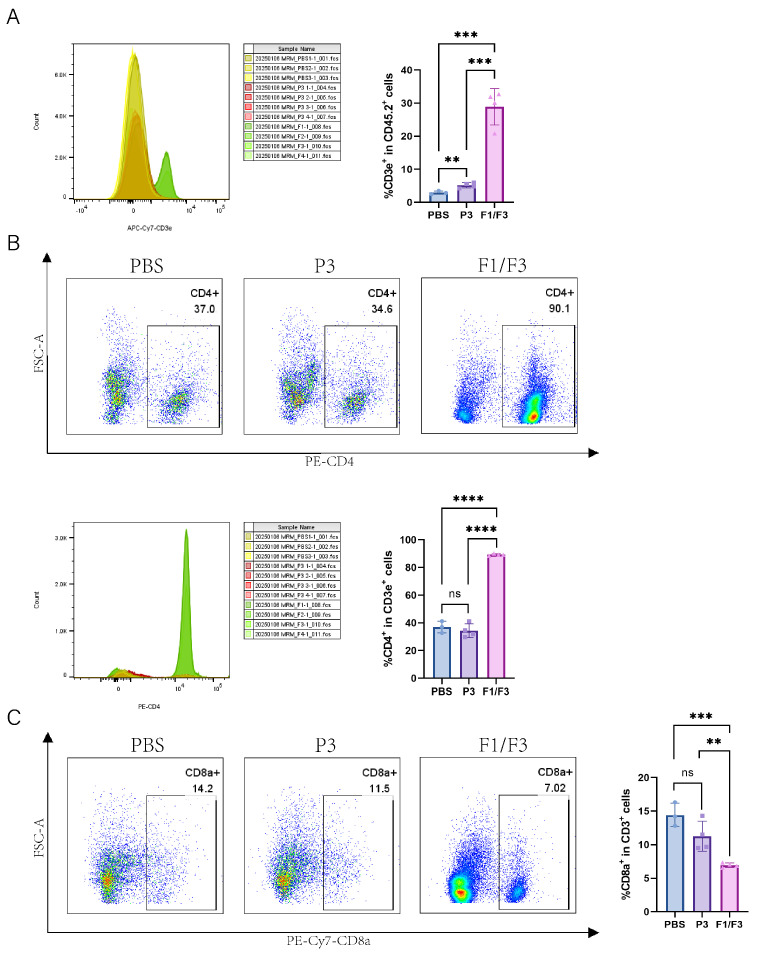
F1/F3 regulating intratumoral T cell responses in 4T-1 tumor-bearing mice. Flow cytometric analysis of T cells in the tumor microenvironment (TME) of PBS-, P3-, and F1/F3-treated 4T-1 tumor-bearing mice, including (**A**) T cells (CD45.2^+^CD3e^+^); (**B**) CD4^+^ T cells (CD45^+^CD3e^+^CD4^+^); (**C**) CD8^+^ T cells (CD45^+^CD3e^+^CD8^+^). PBS group (blue), P3 group (purple), F1/F3 group (pink). Each experiment used 3–6 mice per group. (**A**–**C**) represent single independent experiments, each repeated once. Data are expressed as mean ± SD. ns, not significant; ** *p* < 0.01; *** *p* < 0.001; **** *p* < 0.0001. Statistical analysis was carried out by Student’s *t*-test.

**Figure 4 vaccines-13-00577-f004:**
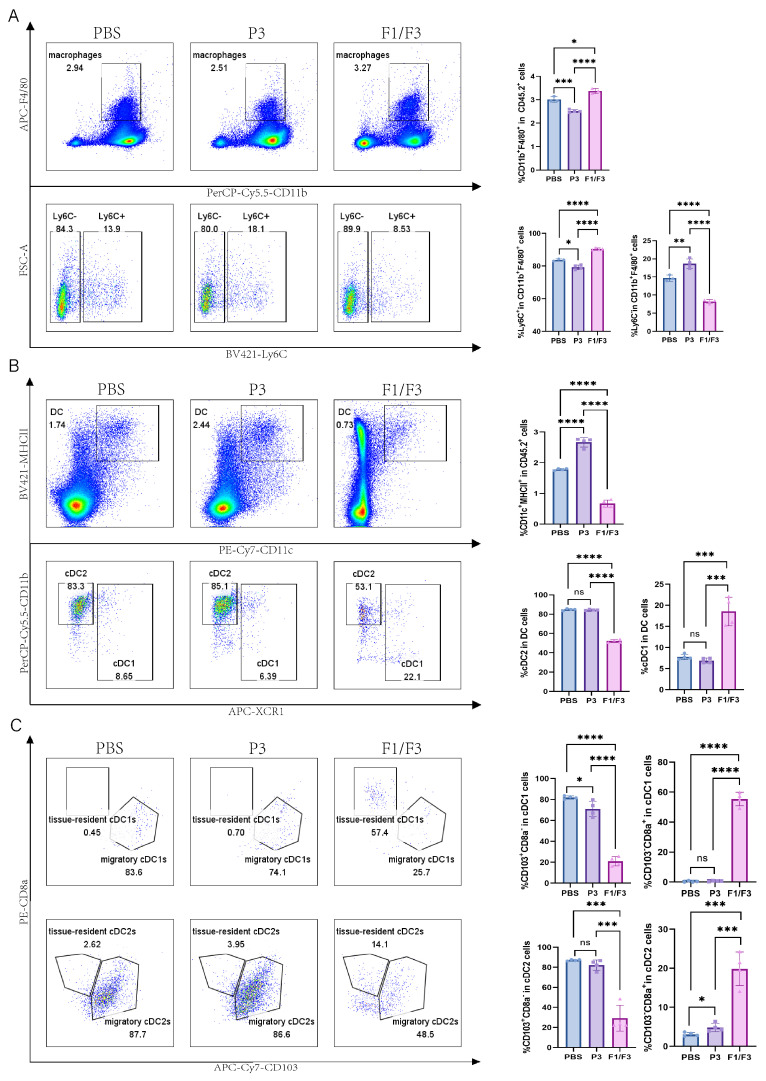
Modulation of intratumoral macrophages and DC subsets in 4T-1 tumor-bearing mice. Flow cytometric analysis of macrophages and DC subsets in the tumor microenvironment (TME) of PBS-, P3-, and F1/F3-treated 4T-1 tumor-bearing mice, including (**A**) macrophages (CD45^+^CD11b^+^F4/80^+^), M1 (CD45^+^CD11b^+^F4/80^+^Ly6C^+^) vs. M2 (CD45^+^CD11b^+^F4/80^+^Ly6C^−^); (**B**) DCs (CD45.2^+^CD11c^+^MHCII^+^), cDC1 (CD45.2^+^CD11c^+^MHCII^+^XCR1^+^), cDC2 (CD45.2^+^CD11c^+^MHCII^+^XCR1^−^CD11b^+^); (**C**) migratory cDC1 (CD45.2^+^CD11c^+^MHCII^+^XCR1^+^CD103^+^CD8a^−^), resident cDC1 (CD45.2^+^CD11c^+^MHCII^+^XCR1^+^CD103^−^CD8a^+^), migratory cDC2 (CD45.2^+^CD11c^+^MHCII^+^XCR1^−^CD11b^+^CD103^+^CD8a^−^), resident cDC2 (CD45.2^+^CD11c^+^MHCII^+^XCR1^−^CD11b^+^CD103^−^CD8a^+^). PBS group (blue), P3 group (purple), F1/F3 group (pink). Each experiment used 3–6 mice per group. (**A**–**C**) represent single independent experiments, each repeated once. Data are expressed as mean ± SD. ns, not significant; * *p* < 0.05; ** *p* < 0.01; *** *p* < 0.001; **** *p* < 0.0001. Statistical analysis was carried out by Student’s *t*-test.

**Figure 5 vaccines-13-00577-f005:**
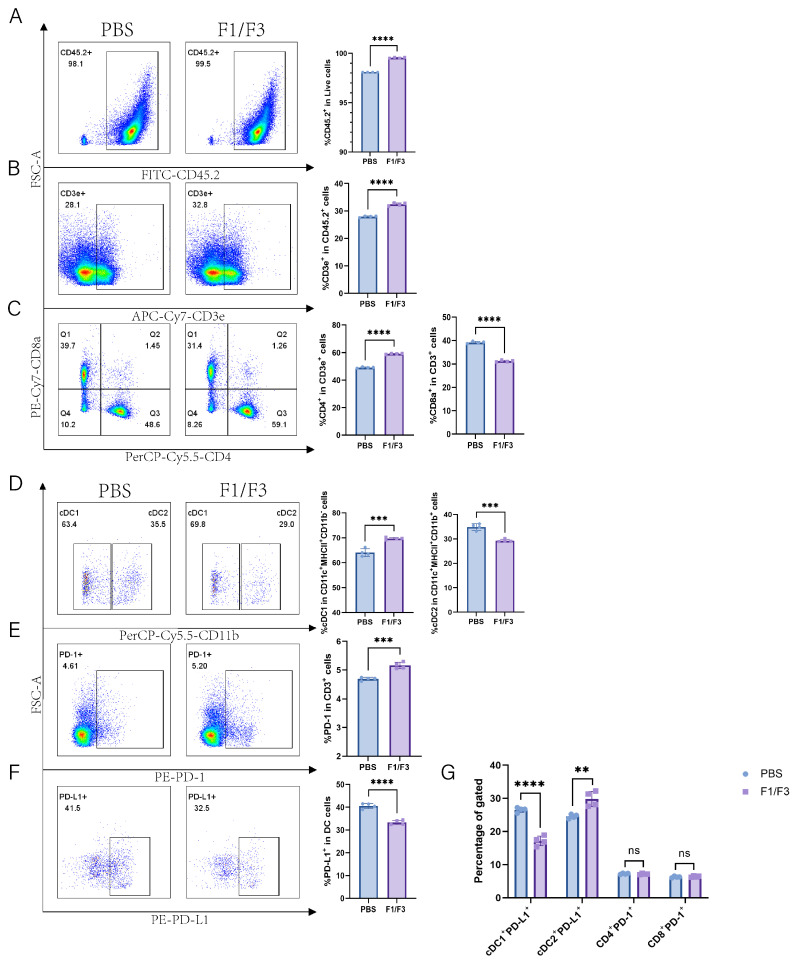
F1/F3 modulates T Cells, DC subsets, and PD-1/PD-L1 expression in the lymph nodes of 4T1 tumor-bearing mice. Flow cytometry was performed on draining lymph nodes from PBS- or F1/F3-treated mice to evaluate immune cells, DC subsets, and related factors: (**A**) CD45.2^+^ cells; (**B**) T cells (CD45.2^+^CD3e^+^); (**C**) CD4^+^ T cells (CD45^+^CD3e^+^CD4^+^), CD8^+^ T cells (CD45^+^CD3e^+^CD8^+^); (**D**) cDC1 (CD45.2^+^Lineage^−^CD11c^+^MHCII^+^CD11b^−^) vs. cDC2 (CD45.2^+^Lineage^−^CD11c^+^MHCII^+^CD11b^+^); (**E**) PD-1 (CD45.2^+^CD3e^+^PD-1^+^); (**F**) PD-L1 (CD45.2^+^Lineage^−^CD11c^+^MHCII^+^PD-L1^+^); (**G**) PD-L1 expression in cDC1 (CD45.2^+^Lineage^−^CD11c^+^MHCII^+^CD11b^−^PD-L1^+^) and cDC2 (CD45.2^+^Lineage^−^CD11c^+^MHCII^+^CD11b^+^PD-L1^+^), as well as PD-1 on CD4^+^ T cells (CD45^+^CD3e^+^CD4^+^PD-1^+^) and CD8^+^ T cells (CD45^+^CD3e^+^CD8^+^PD-1^+^). PBS group (blue), P3 group (purple). Each experiment used 3–6 mice per group. Data in (**A**–**G**) represent single independent experiments, repeated once, shown as mean ± SD or mean ± SEM. ns, not significant; ** *p* < 0.01; *** *p* < 0.001; **** *p* < 0.0001. Statistical analysis was conducted by Student’s *t*-test.

**Figure 6 vaccines-13-00577-f006:**
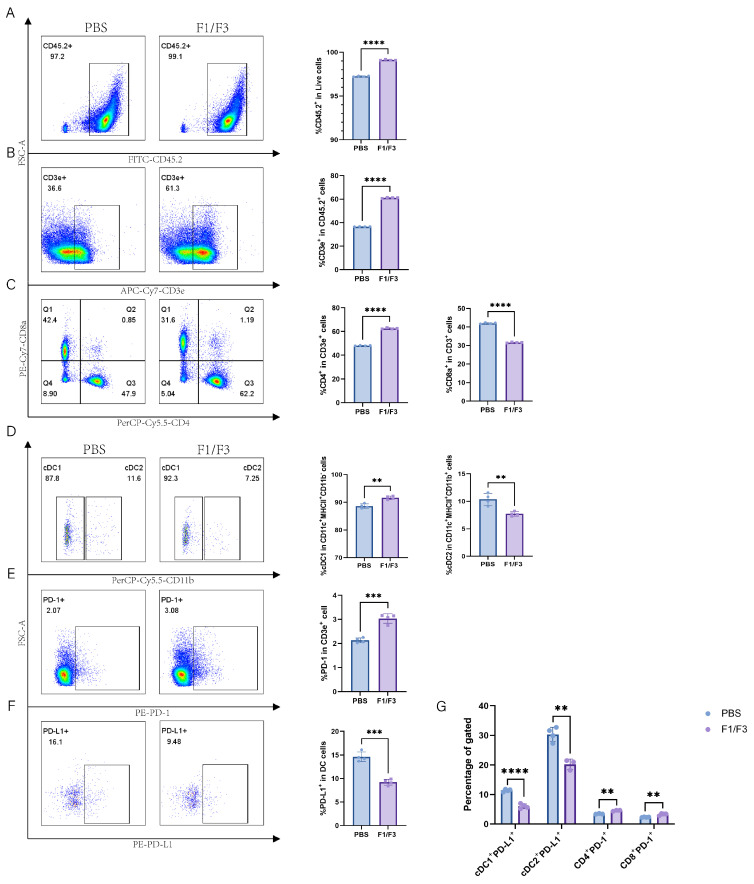
F1/F3 improves the immune profile in non-draining lymph nodes of 4T-1 tumor-bearing Mice. Flow cytometric analysis was performed on non-draining lymph nodes from PBS- or F1/F3-treated mice to examine immune cells, DC subsets, and related markers, including (**A**) CD45.2^+^ cells; (**B**) T cells (CD45.2^+^CD3e^+^); (**C**) CD4^+^ T cells (CD45^+^CD3e^+^CD4^+^), CD8^+^ T cells (CD45^+^CD3e^+^CD8^+^); (**D**) cDC1 (CD45.2^+^Lineage^−^CD11c^+^MHCII^+^CD11b^−^) vs. cDC2 (CD45.2^+^Lineage^−^CD11c^+^MHCII^+^CD11b^+^); (**E**) PD-1 (CD45.2^+^CD3e^+^PD-1^+^); (**F**) PD-L1 (CD45.2^+^Lineage^−^CD11c^+^MHCII^+^PD-L1^+^), and (**G**) PD-L1 expression on cDC1 (CD45.2^+^Lineage^−^CD11c^+^MHCII^+^CD11b^−^PD-L1^+^), cDC2 (CD45.2^+^Lineage^−^CD11c^+^MHCII^+^CD11b^+^PD-L1^+^), as well as PD-1 on CD4^+^ T cells (CD45^+^CD3e^+^CD4^+^PD-1^+^) and CD8^+^ T cells (CD45^+^CD3e^+^CD8^+^PD-1^+^). PBS group (blue), P3 group (purple). Each experiment included 3–6 mice per group. Data in (**A**–**G**) represent a single independent experiment, repeated once. The results are shown as mean ± SD. ns: not significant; ** *p* < 0.01; *** *p* < 0.001; **** *p* < 0.0001. Student’s *t*-test was performed (**A**–**G**).

**Figure 7 vaccines-13-00577-f007:**
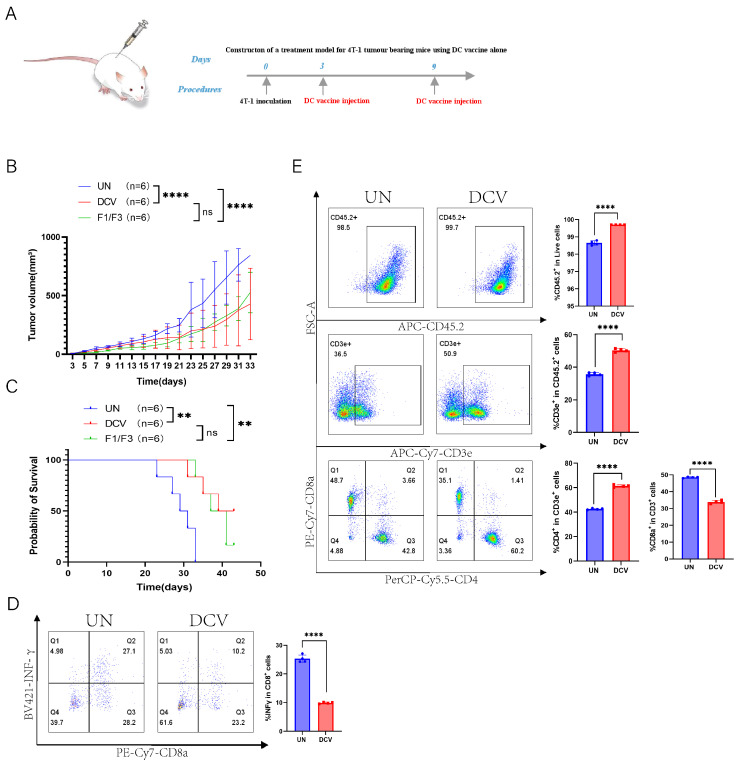
DC vaccine inhibits 4T-1 tumor growth in vivo. (**A**) A 4T-1 tumor-bearing mouse model was established in the lateral flank. Tumor growth (**B**) and survival rates (**C**) were compared among the untreated (UN) group, DC vaccine group (DCV), and F1/F3 group; flow cytometric analysis of draining lymph nodes measured; (**D**) IFN-γ secretion (CD45^+^CD3e^+^CD8^+^IFN-γ^+^) and T-cell subsets, including (**E**) CD45.2^+^ cells, T cells (CD45.2^+^CD3e^+^), CD4^+^ T cells (CD45^+^CD3e^+^CD4^+^), and CD8^+^ T cells (CD45^+^CD3e^+^CD8^+^); UN group (blue), DCV group (red). each group contained 3–6 mice. Data in (**A**–**E**) represent a single independent experiment repeated once, shown as mean ± SD. Two-way ANOVA was used for (**B**), Kaplan–Meier analysis for (**C**), and Student’s *t*-test for (**D**,**E**). ns, not significant; ** *p* < 0.01; **** *p* < 0.0001.

**Figure 8 vaccines-13-00577-f008:**
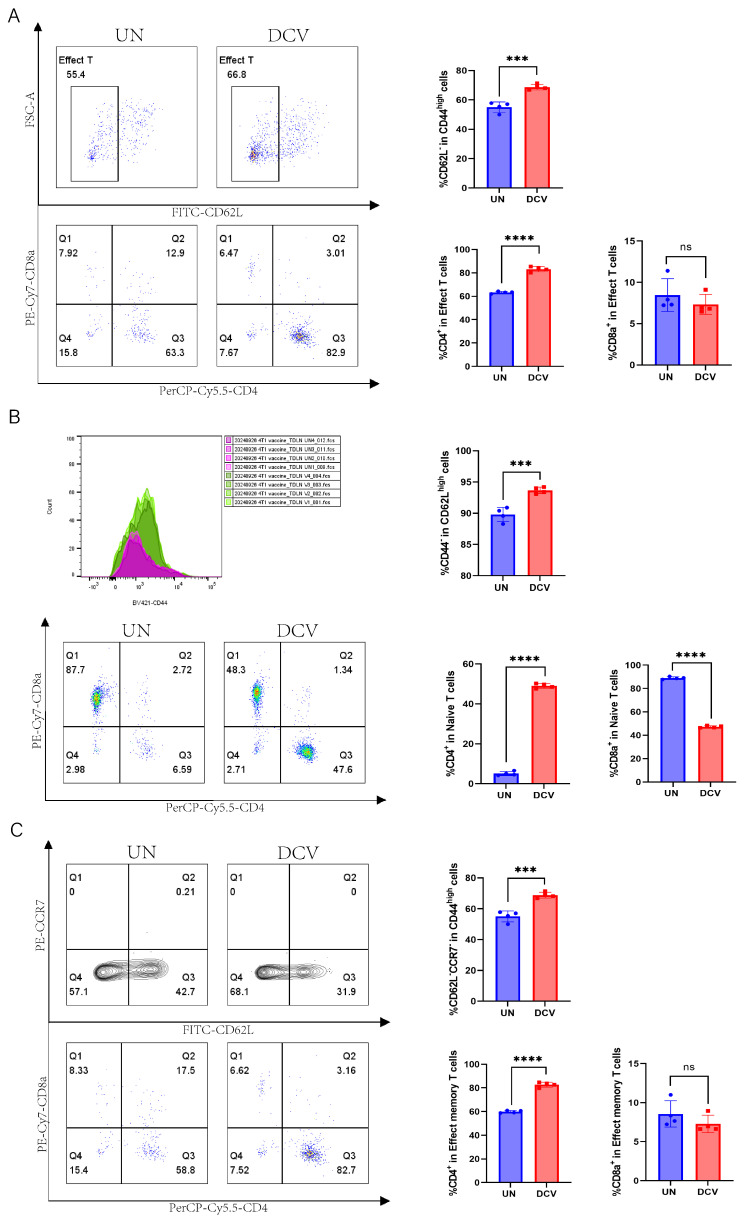
Immunomodulatory effects of DC vaccine on intratumoral T cells in 4T-1 tumor-bearing mice. Flow cytometric analysis of T cell subsets in the draining lymph nodes of the untreated (UN) group and DC vaccine group (DCV), including (**A**) effector T cells (CD45.2^+^CD3e^+^CD44highCD62L^−^), effector CD4^+^ T cells (CD45.2^+^CD3e^+^CD44highCD62L^−^CD4^+^), effector CD8^+^ T cells (CD45.2^+^CD3e^+^CD44highCD62L^−^CD8^+^); (**B**) naïve T cells (CD45.2^+^CD3e^+^CD62LhighCD44^−^), naïve CD4^+^ T cells (CD45.2^+^CD3e^+^CD62LhighCD44^−^CD4^+^), and naïve CD8^+^ T cells (CD45.2^+^CD3e^+^CD62LhighCD44^−^CD8^+^); (**C**) effector memory T cells (CD45.2^+^CD3e^+^CD44highCD62L^−^CCR7^−^), effector memory CD4^+^ T cells (CD45.2^+^CD3e^+^CD44highCD62L^−^CCR7^−^CD4^+^), and effector memory CD8^+^ T cells (CD45.2^+^CD3e^+^CD44highCD62L^−^CCR7^−^CD8^+^). UN group (blue), DCV group (red). Each group contained 3–6 mice. Data in (**A**–**C**) represent a single independent experiment repeated once, shown as mean ± SD. ns, not significant; *** *p* < 0.001; **** *p* < 0.0001. Statistical analysis was conducted by Student’s *t*-test.

**Figure 9 vaccines-13-00577-f009:**
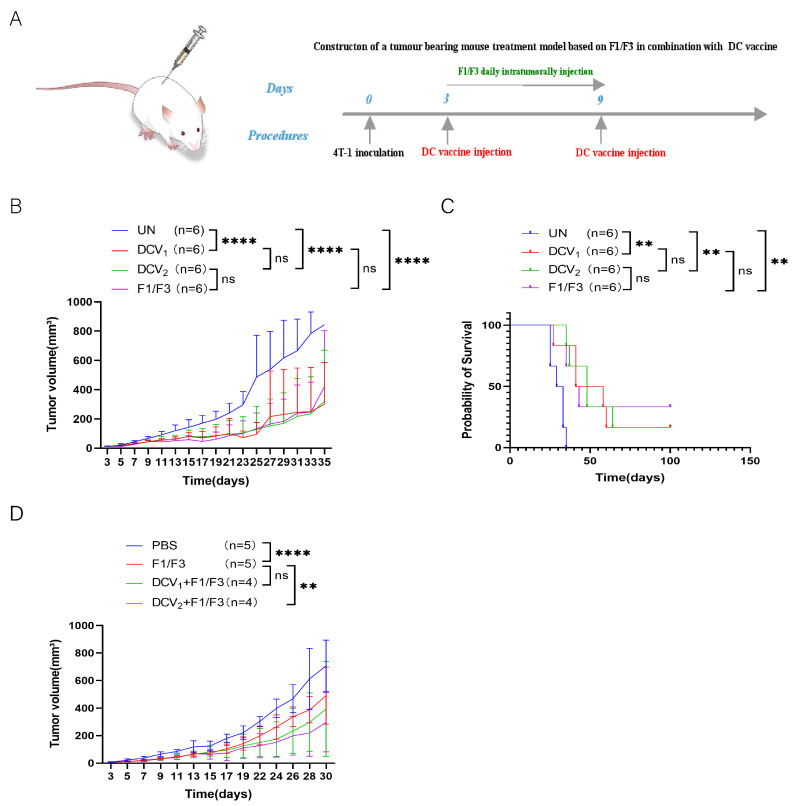
DCV_2_ exhibits superior antitumor activity compared to dCv_1_. (**A**) The 4T-1 tumor-bearing mouse model was established in the lateral flank. Mice received either DCV_1_, DCV_2_, or F1/F3 alone, and (**B**) tumor growth and (**C**) survival rates were recorded. Next, to determine whether DC vaccines could enhance the efficacy of F1/F3, the mice were treated with DCV_1_ + F1/F3 or DCV_2_ + F1/F3 (As shown in (**A**)), and (**D**) tumor growth was measured. Each group contained 3–6 mice. Data in (**A**–**D**) represent single independent experiments, each repeated once. The results are shown as mean ± SD. ns, not significant; ** *p* < 0.01, **** *p* < 0.0001. Statistical significance was evaluated by two-way ANOVA for (**B**,**D**) and Kaplan–Meier survival analysis for (**C**).

**Figure 10 vaccines-13-00577-f010:**
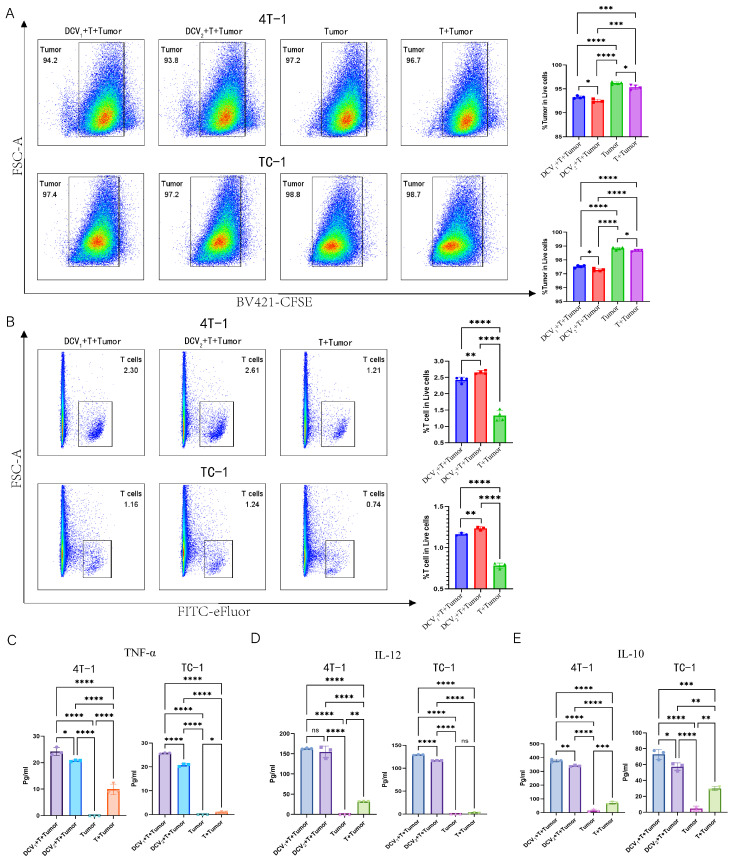
In vitro co-culture of DCV_1_ or DCV_2_ with T Cells and tumor cells. Flow cytometric analysis shows (**A**) that both DCV_1_ and DCV_2_ enhance T cell-mediated cytotoxicity against tumor cells (4T-1/TC-1) (DCV_1_ + T + Tumor (blue), DCV_2_ + T + Tumor (red), Tumor (green), T + Tumor (purple)) and (**B**) promote T-cell proliferation(DCV_1_ + T+Tumor (blue), DCV_2_ + T + Tumor (red), T + Tumor (green)). Using ELISA kits, we quantified immune-related cytokines in the co-culture supernatants, including (**C**) TNF-α, (DCV_1_ + T + Tumor (purple), DCV_2_ + T + Tumor (blue), Tumor (green), T + Tumor (Orange)) (**D**) IL-12, and (**E**) IL-10 (DCV_1_ + T + Tumor (blue), DCV_2_ + T + Tumor (purple), Tumor (pink), T + Tumor (green)). Data in (**A**–**E**) represent a single independent experiment repeated once, shown as mean ± SD. ns, not significant; * *p* < 0.05; ** *p* < 0.01; *** *p* < 0.001; **** *p* < 0.0001. Statistical analysis was conducted by Student’s *t*-test for (**A**,**B**) or one-way ANOVA for (**C**–**E**).

**Table 1 vaccines-13-00577-t001:** Flow cytometry antibodies used in this study.

Antibody	Label	Article Number	Clone	Provider
CD45.2	FITC	11-0454-85	104	eBioscience
CD62L	FITC	11-0621-81	MEL-14	eBioscience
CD86	APC	17-0862-81	GL1	eBioscience
CD45.2	APC	17-0042-83	RM4-5	eBioscience
Ly-6G	APC	17-9668-82	1A8-Ly6g	eBioscience
CD24	APC	101,824	M1/69	Biolegend
CD8a	APC	17-0081-83	53-6.7	eBioscience
CD366	APC	17-5871-82	8B.2C12	eBioscience
F4/80	APC	566,787	T45.2342	BD Bioscience
XCR1	APC	148,205	ZET	Biolegend
F4/80	PE	12-4801-82	BM8	eBioscience
INF-γ	PE	12-7311-82	XMG1.2	eBioscience
NK1.1	PE	12-5941-81	PK136	eBioscience
Granzyme B	PE	12-8898-82	NGZB	eBioscience
CCR7	PE	12-1971-63	4B12	eBioscience
CD223	PE	12-2231-82	eBioC9B7W	eBioscience
PD-1	PE	12-9985-82	J43	eBioscience
PD-L1	PE	124,307	10F.9G2	Biolegend
CD4	PE	557,308	GK1.5	BD Bioscience
CD8a	PE	12-0081-81	53-6.7	eBioscience
CD11b	Percp-cy5.5	45-0112-82	M1/70	eBioscience
CD4	Percp-cy5.5	45-0042-82	RM4-5	eBioscience
CD47	Percp-cy5.5	lot#216680	miap301	lifespan Biosciences
CD8a	Percp-cy5.5	45-0081-82	53-6.7	eBioscience
CD11c	PE-cy7	558,079	HL3	BD Bioscience
CD8a	PE-cy7	552,877	53-6.7	BD Bioscience
PD-L1	PE-cy7	124,314	10F.9G2	Biolegend
CD3e	APC-cy7	557,596	145-2C11	BD Bioscience
B220	APC-cy7	552,094	RA3-6B2	BD Bioscience
CD19	APC-cy7	152,412	1D3/CD19	Biolegend
CD335	APC-cy7	137,647	29A1.4	Biolegend
CD103	APC-Cy7	148,205	ZET	Biolegend
IFN-γ	BV421	563,376	XMG1.2	BD Bioscience
CD47	BV421	127,527	miap301	Biolegend
Ly6C	BV421	562,727	AL-21	BD Bioscience
MHC II	BV421	562,564	M5/114.15.2	BD Bioscience
CD44	BV421	103,039	IM7	Biolegend
Rat IgG1.k. Isotype Control	BV421	562,868	R3-34	BD Bioscience
Fixable Viability stain 510		564,406		BD Bioscience
CD4/CD8 (TIL) MicroBeads		130-116-480		Miltenyi Biotec
anti-CD3e		16-0031-81	145-2C11	eBioscience
Cell Stimulation Cocktail Plus Protein Transport Inhibitor (500×)		00-4975-93		eBioscience

## Data Availability

The original contributions presented in this study are included in this article/Appendix A. Further inquiries can be directed to the corresponding author.

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
