# Peer review of "Synergistic Antitumor Effects of Caerin Peptides and Dendritic Cell Vaccines in a 4T-1 Murine Breast Cancer Model"

_vaccines, 2025, doi:10.3390/vaccines13060577_

Round 1
Reviewer 1 Report
Comments and Suggestions for Authors
Rongmi Mo Et al., investigated the efficacy of combining Caerin peptides (F1/F3) with dendritic cell (DC)- 18
based vaccines in a 4T-1 murine breast cancer model. Using in vitro and in vivo approaches, they reported that the combination of the combination of DCV2 with F1/F3 significantly improved tumor regression and metastasis. They conclude that F1/F3-based DC vaccines, particularly DCV2, synergize with Caerin peptides to be potentially a good therapy in breast cancer.
Concerns
While the paper brings a new information a potential of combined DC and Caerin peptides therapy, several explements were performed either in vitro or in vivo before coming to main finding in the manuscript. I think that the manuscript could benefit from a better organization of data by including some positive data from supplementary information into the main figures and vis versa.
Also, it is not clear why the authors started looking at the apoptosis and pyroptosis (as shown in the figure 1) which was completed ignored in the rest of the manuscript. Not very clear what this information is adding to this manuscript.
In FACS data, the authors used Rat IgG1. k. (Table 1) as isotype control in all staining. A quick verification of Abs table indicated other isotype control are recommended. If so, the authors should be used the right isotype controls or indicate the appropriate controls used for all the FACS staining
Line 175; Please which FACS machine was sued.
Line 219: Is it normal that apoptosis rate in untreated group is around 30%. Isn't that high?
Figure 1: Please revise the legend to figure and give sufficient detail for reader to understand the experiments performed. The IC50 is already commented in the result section. Please revise the legend to figure and give sufficient detail for reader to understand the experiments performed. The IC50 is already commented in the result section.
Figure 2: Did the authors test other route of injection such as IP which will facilitate translation from bench to bed side clinical application.
Line 253: Table 2 is missing in the manuscript, but it is in supplementary data.
Almost in all figure legends of the manuscript, the authors indicate that SEM and SD were used for statistical comparison. They must specify for which graph SD or SEM is used.
Figure 3A; Why the authors choose Dot plots for CD8 T cells expression and histogram for CD4+ T cells. Please show histograms for CD4+ T cells as well.
The authors should also include the gating strategy at least for this figure.
Figure 4E and Fig S2G, Fig.S3A: What is PD-1 expression levels. MFI should be shown in Figs. 4 R-F.
Line 328: Please revise this sentence and correct it.
Figure S5-A: Based on FACS profile, the difference in CD8+ T cells between DCV+T and DCV(TAA) look incremental. Why the authors started Y scale in this figure at 60 %. To avoid biased interpretation of data, the authors should use a scale at 0% as in figS5-B-C.
Fig. 7. If DCV2 have a higher capacity of T cells stimulation compared to DCV1 in vitro, why this is not reflected in tumor suppression in vivo?
It is more relevant, if the authors investigated T cell cytotoxic of T cells from treated mice rather in exploring in vitro killing of tumor cells in vitro
Line 468-469: the authors stated that “the experimental groups (DCV1 + T + Tumor, DCV2 + T + Tumor) exhibited cytotoxic effects against 4T-1 cells, with DCV2 showing beer activity (93.8% vs. 94.2%) compared to DCV1”, this statement is not correct since there is only a minor difference which seems not significant.
The discussion requires some revision to avoid certain redundancies with results section.
A minor English editing is also required.
Comments on the Quality of English Language
A minor English editing is necessay to correct some grammatical and typo mistakes.
Author Response
Dear reviewers and editor,
Thank you very much for reviewing our manuscript, please find below our point-by-point answers to the comments raised by the reviewers and editor.
Reviewer
Comments 1: While the paper brings a new information a potential of combined DC and Caerin peptides therapy, several explements were performed either in vitro or in vivo before coming to main finding in the manuscript. I think that the manuscript could benefit from a better organization of data by including some positive data from supplementary information into the main figures and vis versa.
Response 1: Thank you for thoughtful suggestion. We sincerely appreciate the reviewer's valuable suggestion regarding data reorganization. While we understand the potential benefits of integrating supplementary data into the main figures, we have carefully considered the following aspects:
1.Manuscript length and focus: Moving additional datasets to the main figures might compromise the clarity and conciseness of our core findings.
2.Journal guideLines: We have ensured our current organization complies with the journal's recommendations for supplementary materials.
3.Reader accessibility: All supplementary figures are clearly referenced and directly support the main conclusions.
4.We have relocated selected supplementary figures (Original Figure S2) to the main figures in Figure 6.
Comments 2: Also, it is not clear why the authors started looking at the apoptosis and pyroptosis (as shown in the figure 1) which was completed ignored in the rest of the manuscript. Not very clear what this information is adding to this manuscript.
Response 2: Our research team has previously investigated the capacity of Caerin peptides to induce both apoptosis and pyroptosis in cancer cells (This has been clarified with supporting references in Line 228, Page 7 of the revised manuscript). Therefore, during the initial phase of this project, we incorporated detection assays related to apoptosis and pyroptosis to characterize the effects of F1/F3 peptides on 4T-1 cells. These data provide mechanistic insights that help establish a foundation for interpreting the subsequent observed immune responses in vivo, i.e. the observed higher efficacy of F1/F3 treated 4T-1 cell based DCs vaccine(V2)combined with F1/F3 intratumor injection is better than 4T-1 based DCs vaccine (V1).
Comments 3:In FACS data, the authors used Rat IgG1. k. (Table 1) as isotype control in all staining. A quick verification of Abs table indicated other isotype control are recommended. If so, the authors should be used the right isotype controls or indicate the appropriate controls used for all the FACS staining.
Response 3: Rat IgG1.k was exclusively used as the isotype control for IFN-γ detection. All other antibodies were paired with their appropriate isotype controls (now specified in Table 1 and Methods, Page 7 Line 193 in the revised manuscript).
Comments 4:Line 175; Please which FACS machine was sued.
Response 4: Thank you for your comment. We have specified the flow cytometer model (BD FACSaria II) in Page 4 Line 177 of the Methods section of the revised manusxript.
Comments 5: Line 219: Is it normal that apoptosis rate in untreated group is around 30%. Isn't that high?
Response 5: Thank you for your comment. We acknowledge that the observed apoptosis rate (~30%) in the untreated control group appeared relatively high. After methodological refinements, repeated experiments were conducted, and the updated results are presented in Fig. 1E (Page 8, Line 231).
Comments 6: Please revise the legend to figure and give sufficient detail for reader to understand the experiments performed. The IC50 is already commented in the result section.
Response 6: Thank you for your comment. We have specified the MTT method for IC50 determination in Line 233 on Page 8 of revised manuscript..
Comments 7: Did the authors test other route of injection such as IP which will facilitate translation from bench to bed side clinical application.
Response 7: Thank you for comments. The current study focuses on intratumoral delivery to directly evaluate local immunomodulation, without testing alternative administration routes. We will investigate alternative administration routes in future studies.
Comments 8: Line 253: Table 2 is missing in the manuscript, but it is in supplementary data.
Response 8: Thank you for your comment. We have clearly labeled STable 2 in the Supplementary Materials, specifying the location of pulmonary metastasis data to ensure full data accessibility while maintaining manuscript clarity. (Page 9 Line 264 in revised manuscript).
Comments 9: Almost in all figure legends of the manuscript, the authors indicate that SEM and SD were used for statistical comparison. They must specify for which graph SD or SEM is used.
Response 9: Thank you for your comment. We have explicitly specified in all figure legends whether data are presented as mean ± SD or mean ± SEM.
Comments 10: Figure 3A; Why the authors choose Dot plots for CD8 T cells expression and histogram for CD4+ T cells. Please show histograms for CD4+ T cells as well.
Response 10: Thank you for pointing this out. We have added CD4+ T cell histograms to Figure 3B. (Page 12, Line 300 in revised manuscript).
Comments 11: The authors should also include the gating strategy at least for this figure.
Response 11: We appreciate your comment. We have included the complete gating strategy as Supplementary Figure 10.
Comments 12: Figure 4E and Fig S2G, Fig.S3A: What is PD-1 expression levels. MFI should be shown in Figs. 4 R-F.
Response 12: Thank you for pointing this out. We agree that MFI is important for assessing PD-1 expression intensity. We have included the MFI (Mean Fluorescence Intensity) values of the relevant datasets in Supplementary Figure S11 of the revised manuscript.
Comments 13: Line 328: Please revise this sentence and correct it.
Response 13: Thank you for pointing this out. We agree with this comment. We have revised this section accordingly. (Page 15, Line 337 in revised manuscript)
Comments 14: Figure S5-A: Based on FACS profile, the difference in CD8+ T cells between DCV+T and DCV(TAA) look incremental. Why the authors started Y scale in this figure at 60 %. To avoid biased interpretation of data, the authors should use a scale at 0% as in figS5-B-C.
Response 14: Thank you for your comment. We have revised Figure S5A (now shown in Figure S2A) using a 0% y-axis scale, providing a more balanced visualization of the CD8+ T cell data while preserving the original statistical outcomes.
Comments 15: Fig.7. If DCV2 have a higher capacity of T cells stimulation compared to DCV1 in vitro, why this is not reflected in tumor suppression in vivo?
Response 15: Thanks so much for your thoughtful comment. Previously, we found that a therapeutic HPV16E7 peptide-based vaccine incorporating IL10R antibody stimulate stronger T cell responses in vivo, compared with a same vaccine without IL10R antibody, however, the growth inhibition of TC-1 tumor was slightly influenced[1, 2], the growth inhibition was only significant when the TME was disturbed by F1/F3. The current results are similar to our previous findings. We attribute this phenomenon to the strongly immunosuppressive tumor microenvironment (TME) and the intrinsic complexity of in vivo antigen presentation, as elaborated in the Discussion section (Page 29, Lines 667-670 in the revised manuscript).
Comments 16: It is more relevant, if the authors investigated T cell cytotoxic of T cells from treated mice rather in exploring in vitro killing of tumor cells in vitro.
Response 16: We thank the reviewer for this insightful suggestion. We will incorporate this approach as an important future direction.
Comments 17: Line 468-469: the authors stated that “the experimental groups (DCV1 + T + Tumor, DCV2 + T + Tumor) exhibited cytotoxic effects against 4T-1 cells, with DCV2 showing better activity (93.8% vs. 94.2%) compared to DCV1”, this statement is not correct since there is only a minor difference which seems not significant.
Response 17: Thank you for pointing this out. We agree with this comment. We have revised this section accordingly (Page 25, Line 505-507 in revised manuscript).
Comments 18: The discussion requires some revision to avoid certain redundancies with results section.
Response 18: We sincerely appreciate the reviewer's valuable suggestion regarding the redundancy between the Results and Discussion sections. We have revised the Discussion section, specifically streamlining the descriptions on Page 27, Line 543 and Line 546, and deleting the results descriptions on Page 27, Line 581; Page 28, Line 601; and Page 29, Line 667.
Comments 19: A minor English editing is also required.
Response 19: We appreciate the reviewer's feedback regarding English language editing. We has carefully revised the manuscript, and the manuscript has been edited by a professional English writer to correct grammatical errors, awkward phrasing, or unclear expressions.
- Ni G, Zhang L, Yang X, Li H, Ma B, Walton S, Wu X, Yuan J, Wang T, Liu X: Targeting interleukin-10 signalling for cancer immunotherapy, a promising and complicated task. Hum Vaccin Immunother 2020, 16(10):2328-2332.
- Ni G, Yang X, Li J, Wu X, Liu Y, Li H, Chen S, Fogarty CE, Frazer IH, Chen Get al: Intratumoral injection of caerin 1.1 and 1.9 peptides increases the efficacy of vaccinated TC-1 tumor-bearing mice with PD-1 blockade by modulating macrophage heterogeneity and the activation of CD8(+) T cells in the tumor microenvironment. Clin Transl Immunology 2021, 10(8):e1335.
Reviewer 2 Report
Comments and Suggestions for Authors
This study explores the impact of Caerin 1.1/1.9 on the proliferation and apoptosis of the 4T-1 mouse breast cancer cell line. The researchers found that Caerin 1.1/1.9 significantly inhibited cell growth and induced apoptosis, exhibiting characteristics similar to pyroptosis. Additionally, Caerin1.1/1.9 reduced tumor growth and lung metastasis. Flow cytometry analysis revealed that Caerin 1.1/1.9 treatment increased CD4⁺ T cell infiltration while decreasing CD8⁺ T cells, suggesting a mechanism where CD4⁺ T cells support CD8⁺ T cell functions. The treatment also led to an increase in M2 macrophages and a decrease in pro-inflammatory M1 macrophages, which tumor-derived chemokines could influence. Notably, discrepancies in immune responses between different cancer models were observed, attributed to variations in tumor immunogenicity and the genetic backgrounds of the mouse strains used. F1/F3 also promoted the upregulation of cross-presenting dendritic cells, enhancing potential immune activation. Lastly, Caerin1.1/1.9 treatment affected the lymphatic system by increasing T cells in both tumor-draining and non-draining lymph nodes, suggesting its relevance in cancer immunotherapy.
Comments:
- Why was the intratumoral route chosen for administering Caerin 1.1 and 1.9 compounds? Are these peptides specifically sequestered in cancer cells compared to normal cells? If they are not, please address the potential toxic side effects of Caerin 1.1 and 1.9 peptides.
- Please report the IC50 values along with the standard deviation (SD) or standard error of the mean (SEM).
- How were the apoptosis rates calculated in Figure 1E? The proportional differences displayed in the bar chart do not align with the differences observed in the flow cytometry data.
- To demonstrate that dendritic cells (DCs) loaded with TC-1 cells promote T-cell proliferation, the authors calculated the percentages of CD4 and CD8 T cells through flow cytometry in the mixed cell culture. However, flow cytometric analysis may yield false positive results for T-cell proliferation due to disproportionate non-T-cell death in the mixed culture. Ideally, in this type of assay, T cells should be labeled with CFSE dye, allowing for the demonstration of increased proliferation via dilution of the CFSE dye in specific T-cell populations. A similar method and analysis were presented in Figure 7B.
- Please correct the typo (TBD) in line 189.
- Please increase the font size of all the figures. Without enlarging each figure, it is difficult to read the axis text.
Author Response
Dear reviewers and editor,
Thank you very much for reviewing our manuscript, please find below our point-by-point answers to the comments raised by the reviewers and editor.
Reviewer
Comments 1: Why was the intratumoral route chosen for administering Caerin 1.1 and 1.9 compounds? Are these peptides specifically sequestered in cancer cells compared to normal cells? If they are not, 2. please address the potential toxic side effects of Caerin 1.1 and 1.9 peptides.
Response 1: We sincerely appreciate the reviewer's question regarding the administration route and safety profile of Caerin peptides. Based on our previous studies (Ni et al., Front Cell Dev Biol 2020; Yang et al., Cancers 2022), we selected intratumoral administration because this route maximizes local peptide concentration while minimizing systemic exposure, as demonstrated in our pharmacokinetic studies (Yang et al., Evid Based Complement Alternat Med 2022); our toxicity evaluations in rodent models showed excellent safety profiles at therapeutic doses, with only transient, mild inflammation at injection sites; while the peptides exhibit broad-spectrum activity, their membranolytic effects show preferential activity against cancer cells due to differences in membrane composition between normal and cancer cells.
Comments 2: Please report the IC50 values along with the standard deviation (SD) or standard error of the mean (SEM).
Response 2: We have now included the standard deviation (SD) values for all reported IC50 measurements in the revised manuscript.(Page 7, Line 218 in the revised manuscript)
Comments 3: How were the apoptosis rates calculated in Figure 1E? The proportional differences displayed in the bar chart do not align with the differences observed in the flow cytometry data.
Response 3: In our study, the apoptosis rates presented in Figure 1E were calculated as the average of early apoptotic cells (Annexin V+/PI-, Q3 quadrant) and late apoptotic cells (Annexin V+/PI+, Q2 quadrant) from the flow cytometry analysis. Flowcytometry profile of representative results among different groups were selected.
Comments 4: To demonstrate that dendritic cells (DCs) loaded with TC-1 cells promote T-cell proliferation, the authors calculated the percentages of CD4 and CD8 T cells through flow cytometry in the mixed cell culture. However, flow cytometric analysis may yield false positive results for T-cell proliferation due to disproportionate non-T-cell death in the mixed culture. Ideally, in this type of assay, T cells should be labeled with CFSE dye, allowing for the demonstration of increased proliferation via dilution of the CFSE dye in specific T-cell populations. A similar method and analysis were presented in Figure 7B.
Response 4: We sincerely appreciate the reviewer's insightful suggestion regarding the T-cell proliferation assay methodology. We fully acknowledge the importance of using CFSE labeling for accurately quantifying T-cell proliferation, as it provides direct evidence of cell division through dye dilution. In Figure S4C, the expression level of IFNγ can also highlight the role of DCs loaded with TC-1 cells in promoting T cell function.
Comments 5: Please correct the typo (TBD) in Line 189.
Response 5: Thank you for pointing this out. TBD represents the brand name of the reagent, not an abbreviation, and has been updated to 'TBD science'. (Page 7, Line 197)
Comments 6: Please increase the font size of all the figures. Without enlarging each figure, it is difficult to read the axis text.
Response 6: We have uniformly scaled up all figures and proportionally enlarged the numerical font sizes to enhance readability.
Reviewer 3 Report
Comments and Suggestions for Authors
The manuscript titled "Synergistic Antitumor Effects of Caerin Peptides and Dendritic Cell Vaccines in a 4T-1 Murine Breast Cancer Model" by Mo et al. presents an interesting and timely investigation into the combination of Caerin peptides (F1/F3) and dendritic cell (DC) vaccines for enhancing antitumor immunity in a murine breast cancer model. The study is thorough in its experimental design, and the use of both in vitro and in vivo analyses supports the validity of the findings. However, there are some points in the article to which the authors should pay attention so that the manuscript can be published in the journal.
- The rationale behind combining Caerin peptides with DC vaccines is scientifically sound, but the novelty of this combination should be better emphasized in the introduction. How does this approach compare to other combination therapies used in breast cancer models?
- The flow cytometry data are extensive, but histograms are too small and unreadable (e.g., Fig. 3, 4, 5, 7). It would be better to separate these figures and enlarge them.
- In the text the authors write “Tumor”, but in Figure 7 they write “Tumour”. This should be corrected to maintain stylistic uniformity.
- The results suggest that F1/F3 peptides influence immune cell infiltration, especially promoting CD4+ T cell and cDC1 infiltration. However, the mechanism underlying this selective recruitment remains speculative. Have the authors examined chemokine expression patterns in treated tumors?
Author Response
Dear reviewers and editor,
Thank you very much for reviewing our manuscript, please find below our point-by-point answers to the comments raised by the reviewers and editor.
Reviewer
Comments 1: The rationale behind combining Caerin peptides with DC vaccines is scientifically sound, but the novelty of this combination should be better emphasized in the introduction. How does this approach compare to other combination therapies used in breast cancer models?
Response 1: We highly appreciate the comment of the reviewer. We have expanded the Introduction (Page 2, Lines 93-101 in revised manuscript) to better highlight the novelty of this combination approach. We also include current status of other combination therapies in breast cancer model (Page 16, Lines 361).
Comments 2: The flow cytometry data are extensive, but histograms are too small and unreadable (e.g., Fig. 3, 4, 5, 7). It would be better to separate these figures and enlarge them.
Response 2: We have uniformly scaled up all figures and proportionally enlarged the numerical font sizes to enhance readability.
Comments 3: In the text the authors write “Tumor”, but in Figure 7 they write “Tumour”. This should be corrected to maintain stylistic uniformity.
Response 3: We apologize for this oversight and have standardized all instances to "tumor" in Figure 7(now shown in Figure 10).
Comments 4: The results suggest that F1/F3 peptides influence immune cell infiltration, especially promoting CD4 T cell and cDC1 infiltration. However, the mechanism underlying this selective recruitment remains speculative. Have the authors examined chemokine expression patterns in treated tumors?
Response 4: Thank you for your suggestion. Why CD4+T cells and cDC1 are specifically attracted to 4T-1 tumor remain elusive, chemokine level changes following the treatment of F1/F3 will be the focus of our upcoming study. Add one or two sentences in discussion in the revised manuscript.(Page 28, Lines 590-592)